# Adaptive Utilization of Low-Rank Adaptation via Conditioned Gating

**Guang Yang** [* 1 2] **Changhao Guan** [* 1 2] **Chao Huang** [1 2] **Yufeng Chen** [1 2] **Kaiyu Huang** [1 2]

## Abstract

Low-Rank Adaptation (LoRA) achieves parameter-efficient fine-tuning by constraining model updates to a low-rank subspace and has been widely used in practice. However, LoRA typically employs a shared low-rank update across tokens, which limits its ability to fully exploit the adaptation subspace for tokens from different sequences. To address this issue, we propose an adaptive utilization of Low-Rank Adaptation (U-LoRA), which employs conditioned gating to explicitly learn effective token-level utilization of the limited low-rank adaptation subspace. Specifically, U-LoRA generates utilization coefficients along low-rank directions for each token and jointly coordinates and constrains them using sequence-level contextual information, thereby inducing more consistent adaptive patterns within a sentence. To further enhance training stability, we introduce a bias-corrected exponential moving average (EMA) historical prior that calibrates utilization signals across optimization steps, suppressing noise caused by batch-to-batch fluctuations. The effectiveness of our method arises from a better utilization of the existing low-rank subspace via input-conditioned strategies, rather than from expanding the subspace. Experiments on mathematical reasoning and natural language understanding benchmarks demonstrate that U-LoRA achieves competitive performance under comparable parameter budgets when with strong LoRA baselines and recent variants.

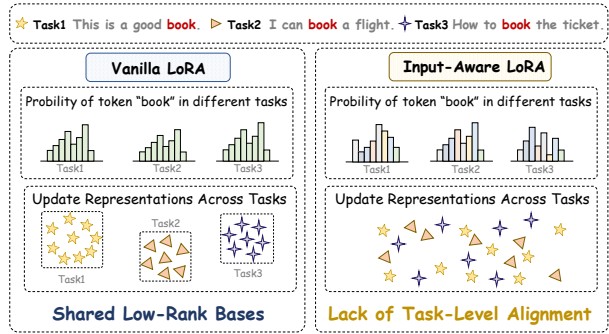

*Figure 1.* The same token may require different low-rank update directions under different task contexts. Vanilla LoRA reuses similar directions due to shared low-rank bases, while input-aware variants diversify directions but may lack task-consistent utilization.

## 1. Introduction

Parameter-efficient fine-tuning (PEFT) is widely used to adapt large pretrained language models to downstream tasks with lower training costs (Lester et al., 2021; He et al., 2021; Fu et al., 2023; Han et al., 2024; Mao et al., 2024). Among PEFT methods, Low-Rank Adaptation (LoRA) (Hu et al., 2022) achieves efficient adaptation by freezing the backbone parameters and introducing only a small number of low-rank trainable parameters, and has been widely adopted in practical applications (Xu et al., 2023; Dettmers et al., 2023).

However, in vanilla LoRA, parameter updates are constrained to a fixed-rank low-dimensional subspace that is shared among all input contexts (Valipour et al., 2023; Dou et al., 2024; Guan et al., 2025; Gao et al., 2025). It leads to interference among updates corresponding to different semantic contexts, potentially degrading performance in tasks that require fine-grained and context-sensitive behavior (Deshpande et al., 2023; Zhang et al., 2025a; Huang et al., 2025a). As illustrated in Figure 1(left), the same token (e.g., *book*) can correspond to different semantic roles across tasks, yet vanilla LoRA tends to reuse highly similar low-rank update directions due to its shared low-rank bases, which limits its ability to disentangle task-specific signals in a fixed subspace. To mitigate this issue, recent work has explored an input-aware LoRA adaptation paradigm, which introduces input-conditioned routing or token-level

---

[*]Equal contribution [1]School of Computer Science and Technology, Beijing Jiaotong University, Beijing 100044, China [2]Key Laboratory of Big Data & Artificial Intelligence in Transportation (Beijing Jiaotong University), Ministry of Education. Correspondence to: Kaiyu Huang <kyhuang@bjtu.edu.cn>.

activation modulation to dynamically allocate low-rank components based on different inputs, thereby enhancing the model's adaptation flexibility (e.g., LoRA-MoE (Luo et al., 2024), HydraLoRA (Tian et al., 2024), and TopLoRA (Li et al., 2025)). In practice, the utilization decisions of these methods often rely on local and relatively simple signals, making it difficult to ensure task-level consistency under complex contextual dependencies. As shown on the right side of Figure 1, although input-aware LoRA methods enable more flexible low-rank direction selection at the token level, these selections are typically made independently, resulting in dispersed and uncoordinated low-rank representation updates within the same task or sentence, which in turn undermines alignment at the sequence and task levels. As a result, effective LoRA methods need to mitigate the limitation that the shared low-rank subspace imposes on expressive capacity, while simultaneously preserving the independence of representations across tasks.

Motivated by these limitations, we propose an adaptive utilization of Low-Rank Adaptation (U-LoRA), which explicitly models the utilization of the limited low-rank capacity. Building upon the parameterization of vanilla LoRA, U-LoRA learns an input-conditioned utilization strategy over low-rank directions, enabling the adaptation capacity to be selectively allocated according to the input rather than uniformly shared. In particular, U-LoRA introduces a token-conditioned mechanism that assigns usage coefficients to low-rank directions. For each token, U-LoRA generates a set of scalar coefficients that modulate the contribution of individual low-rank directions. These coefficients dynamically reweight the shared low-rank basis, allowing different tokens and crucially different semantic instantiations of the same token to activate the low-rank directions with varying strengths. To avoid making utilization decisions independently across tokens, we aggregate these signals at the sequence level to encourage consistent utilization patterns within a sentence. In addition, frequent input-conditioned learning may cause utilization decisions to fluctuate more during optimization, thereby affecting training stability. To address this issue, we introduce a lightweight historical prior that calibrates utilization signals across optimization steps, improving stability during fine-tuning. We evaluate U-LoRA on mathematical reasoning and GLUE natural language understanding benchmarks across diverse architectures and scales, including decoder-only Large Language Models (LLMs) (Gemma, LLaMA, Qwen) and encoder-based models (RoBERTa-Base and RoBERTa-Large). Results show that under comparable parameter budgets, U-LoRA delivers consistent gains across benchmarks and reliably outperforms representative LoRA baselines and their variants. In summary, our contributions are threefold:

- We rethink LoRA-style PEFT from the perspective of low-rank utilization. Under rank-constrained conditions, adaptation signals from different token positions share the same low-rank directions, motivating an explicit utilization strategy that prioritizes the limited capacity towards task-critical positions.

- We propose U-LoRA that learns token-level utilization coefficients over low-rank directions and coordinates them with a sequence-level aggregation module, enabling context-consistent utilization within a sentence while preserving LoRA's parameter efficiency.

- To improve robustness during fine-tuning, we introduce a lightweight historical prior that calibrates utilization signals across optimization steps, which consistently outperforms strong LoRA baselines and recent variants under comparable parameter budgets.

## 2. Related Work

**Parameter-Efficient Fine-Tuning.** PEFT aims to efficiently adapt large pretrained models to downstream tasks. Early PEFT approaches typically freeze most pretrained parameters and introduce lightweight trainable components, such as adapters and their variants (Houlsby et al., 2019; Pfeiffer et al., 2021), as well as prompt tuning and prefix-style tuning (Liu et al., 2023). LoRA (Hu et al., 2022) further advances the reparameterization line by constraining weight updates to a low-rank form, and has become one of the most widely used PEFT baselines due to its simplicity and strong engineering practicality. A large body of follow-up work improves LoRA from the perspectives of capacity and optimization. Capacity-oriented variants increase effective expressiveness by allocating and transforming low-rank structures, such as AdaLoRA (Zhang et al., 2023) and structured low-rank variants including MELoRA (Ren et al., 2024), HiRA (Huang et al., 2025b), and KronA (Edalati et al., 2025). Optimization-oriented variants enhance learnability and training stability through reparameterization and improved initialization, such as DoRA (Liu et al., 2024), PiSSA (Meng et al., 2024) and LoRA-GA (Wang et al., 2024). Despite the advances, these approaches primarily refine the low-rank structure or the training procedure, while the research question of how low-rank directions should be *utilized* under varying inputs and contexts remains unexplored.

**Context-Aware Low-Rank Utilization.** Recent work introduces input-conditioned mechanisms to enable more fine-grained low-rank adaptation(Dou et al., 2024; Zhang et al., 2025b). LoRA-XS (Bałazy et al., 2024) and LoRA-SB (Ponkshe et al., 2024) highlight the role of input–output projections by decomposing LoRA into subspaces and learning only key projection components under a tighter parameter budget. MoE-style LoRA variants, such as MoELoRA (Luo et al., 2024) and HydraLoRA (Tian et al.,

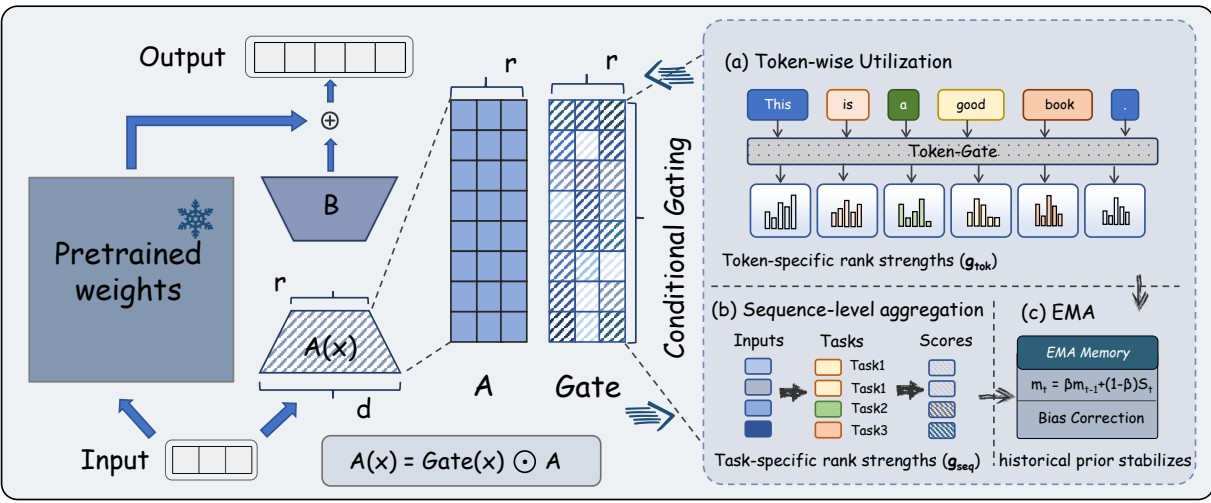

Figure 2. U-LoRA augments standard LoRA with token-wise, input-conditioned gating to modulate low-rank updates. Token features are projected into a rank-aligned space and combined with sequence-level context to coordinate utilization. During training, a lightweight historical prior stabilizes the gating signals across optimization steps, while the pretrained backbone remains frozen.

2024), allow different inputs to select or combine multiple low-rank components to increase adaptivity. TopLoRA (Li et al., 2025) further introduces token-wise diagonal modulation to obtain token-dependent input–output projections. Despite enhancing input-conditioned low-rank adaptation, these methods still lack explicit modeling of low-rank utilization within a fixed subspace to ensure task-consistent behavior, and how to maintain such utilization stably throughout training remains an open question.

## 3. Methodology

### 3.1. The Overview

As illustrated in Figure 2, we integrate an adaptive utilization into vanilla LoRA, which consists of the token-wise conditional gating and sequence-level aggregation module. Specifically, U-LoRA predicts a low-rank gating matrix for each token and modulates the low-rank projection components element-wise, while incorporating sequence-level aggregation to coordinate and constrain token-level gating. To improve training stability, U-LoRA adopts a bias-corrected exponential moving average (EMA) historical prior to calibrate utilization signals across optimization steps and suppress batch-to-batch fluctuations.

### 3.2. Parameterization with Input-conditioned Gating

We consider a pretrained linear layer[1] with weight $W_0 \in \mathbb{R}^{d_{\text{out}} \times d_{\text{in}}}$ and input $x \in \mathbb{R}^{d_{\text{in}}}$. Following LoRA, we freeze

---

[1]Here, *linear layer* broadly refers to projection matrices in Transformers, such as $W_Q, W_K, W_V, W_O$ in attention, as well as projection layers in the feed-forward network.

$W_0$ and restrict the weight update to a rank-$r$ decomposition:

$$W = W_0 + \Delta W, \qquad \Delta W = BA, \qquad (1)$$

where $A \in \mathbb{R}^{r \times d_{\text{in}}}$ and $B \in \mathbb{R}^{d_{\text{out}} \times r}$ with $r \ll \min(d_{\text{out}}, d_{\text{in}})$. The resulting low-rank update is

$$\Delta h = BAx. \qquad (2)$$

Here, $A$ projects the input into a low-rank space, while $B$ maps the low-rank representation back to the output space.

From a parameterization perspective, vanilla LoRA assumes that the down-projection matrix $A$ is constant across all inputs. We generalize this formulation by allowing the effective down-projection to depend on the input, while keeping the overall low-rank structure unchanged. Formally, we replace the constant matrix $A$ with an input-conditioned matrix-valued function $A(x)$ and write the update as

$$\Delta h = B\big(A(x)x\big). \qquad (3)$$

To ensure that the update remains within the same rank-$r$ subspace, we parameterize $A(x)$ as an element-wise modulation of $A$:

$$A(x) = G(x) \odot A, \qquad (4)$$

where $G(x)$ has the same shape as $A$. Under this parameterization, $A$ and $B$ are learned as shared low-rank bases, while the gating function $G(x)$ determines the utilization of these bases for a given input. During training, we freeze the pretrained weights $W_0$ and optimize the adaptation parameters $\{A, B\}$ together with the gating function $G(\cdot)$. When $G(x) \equiv \mathbf{1}$, the formulation reduces to vanilla LoRA.

### 3.3. Token-wise Utilization and Sequence-level Aggregation

After establishing the input-conditioned parameterization of low-rank adaptation, we combine token-wise and sequence-level signals to construct token-specific gating representations, which are then used to parameterize the input-conditioned gating function $G(X)$ for modulating the shared low-rank adaptation. Let $X = [x_1, \ldots, x_s]$ denote the sequence of input representations, where $x_t \in \mathbb{R}^{d_{\text{in}}}$. We first project the input into a rank-aligned feature space for utilization modeling:

$$F = \text{RMSNorm}(XW_{\text{proj}}), \tag{5}$$

where $W_{\text{proj}} \in \mathbb{R}^{d_{\text{in}} \times r}$ and $f_t \in \mathbb{R}^r$ denotes the $t$-th row of $F$. Based on $\{f_t\}_{t=1}^s$, we construct a sequence-level directional summary:

$$s = \sum_{t=1}^{s} \text{softmax}(f_t) \odot f_t, \tag{6}$$

where the softmax is applied along the rank dimension. This summary captures sequence-level preferences over low-rank directions and serves as a global conditioning signal. In the decoder-only architecture, the sequence-level aggregation is computed with a causal mask, ensuring that each token only depends on the current prefix.

We then combine token-level and sequence-level signals in an additive manner to form an intermediate gated representation. For each token $t$, we define

$$z_t = f_t + g_{\text{tok}}(f_t) + g_{\text{seq}}(s), \tag{7}$$

where $g_{\text{tok}}$ and $g_{\text{seq}}$ ($\mathbb{R}^r \to \mathbb{R}^r$) are learnable linear mappings. Under this formulation, $z_t$ incorporates both token-conditioned and sequence-conditioned modulation signals and is used to produce the token-specific gate $G_t(X)$.

### 3.4. Stabilizing Utilization with a Historical Prior

During training, the gating signal is induced from mini-batch statistics and its distribution may vary across optimization steps. To calibrate the gating signal across steps, we introduce a bias-corrected exponential moving average historical prior and apply it in the gate update process.

At training step $t$, suppose the mini-batch contains $B$ sequences. For each sequence $i$, we compute the sequence-level directional summary $s^{(i)} \in \mathbb{R}^r$ as defined in Section 3.3, and take the batch mean

$$\bar{s}_t = \frac{1}{B} \sum_{i=1}^{B} s^{(i)}. \tag{8}$$

We maintain an EMA historical statistic $m_t \in \mathbb{R}^r$ (initialized as $m_0 = \mathbf{0}$) and update it as

$$m_t = \beta m_{t-1} + (1-\beta)\bar{s}_t, \qquad \beta \in [0,1), \tag{9}$$

with the standard bias correction

$$\hat{m}_t = \frac{m_t}{1 - \beta^t}. \tag{10}$$

We use $\hat{m}_t$ as a historical prior and directly calibrate the intermediate gating representation by defining the token-specific gate

$$G_t(X) = z_t + \hat{m}_t, \qquad t = 1, \ldots, s, \tag{11}$$

where $\hat{m}_t$ is broadcast across the sequence dimension to each token. This historical prior provides cross-step smoothing of the gating signal, reducing gate noise induced by batch-to-batch fluctuations. At inference time, the model leverages the accumulated historical information to regulate the gating signal without performing any further updates.

Finally, the forward computation for token $t$ is given by

$$h_t = W_0 x_t + \gamma \cdot B\big((G_t(X) \odot A)x_t\big), \tag{12}$$

where $\gamma$ is the scaling factor for the low-rank branch. In addition, this historical prior is introduced as a training-time regularization mechanism to stabilize utilization learning. During inference, the historical prior is frozen, and no additional computation is required.

## 4. Experiments

### 4.1. Experimental Setup

**Datasets and Models.** We evaluate U-LoRA on reasoning and natural language understanding (NLU) tasks, covering both encoder-only and decoder-only architectures across multiple model scales. For reasoning tasks, we use the mathematical reasoning benchmark compiled in LLM-Adapters (Hu et al., 2023), including AddSub (Hosseini et al., 2014), MultiArith (Roy & Roth, 2015), SingleEq (Koncel-Kedziorski et al., 2015), GSM8K (Cobbe et al., 2021), AQuA (Ling et al., 2017), and SVAMP (Patel et al., 2021), and evaluate on four LLMs with different model families: Gemma-7B (Team et al., 2024), LLaMA-3-8B (Dubey et al., 2024), Qwen3-8B (Yang et al., 2025) and Qwen2.5-14B (Yang et al., 2024). For NLU tasks, we adopt the GLUE benchmark (Wang et al., 2018) which consists of eight tasks, and report results with RoBERTa-Base and RoBERTa-Large (Liu et al., 2019). Detailed evaluation metrics, data splits, and task-specific training details are provided in Appendix A.

**Baseline Methods.** We compare U-LoRA with vanilla LoRA and a set of representative LoRA variants that follow different enhancement strategies: (1) **LoRA** performs parameter-efficient adaptation via low-rank matrix decomposition on frozen weights, (2) **DoRA** (Liu et al., 2024)

*Table 1.* Mathematical reasoning benchmark results of different parameter-efficient fine-tuning methods on multiple LLMs. The highest average precision is bolded, and the second-highest is underlined.

| Model | Method | #Params | AddSub | MultiArith | SingleEq | GSM8K | AQuA | SVAMP | Avg |
|---|---|---|---|---|---|---|---|---|---|
| Gemma-7B | LoRA ($r$=8) | 4.82M | 87.59 | 90.33 | 89.76 | 56.10 | 29.13 | 75.70 | 71.44 |
| | LoRA ($r$=16) | 9.63M | 86.84 | 92.83 | 89.57 | 58.15 | 30.71 | 74.90 | 72.17 |
| | LoRA ($r$=32) | 19.30M | 86.58 | 91.50 | 91.93 | 58.45 | 32.28 | 75.50 | 72.71 |
| | DoRA ($r$=16) | 9.98M | 87.59 | 94.17 | 91.34 | 58.68 | 27.95 | 75.80 | 72.59 |
| | MELoRA ($r$=16) | 9.63M | 87.26 | 92.22 | 91.01 | 59.49 | 32.15 | 74.97 | 72.85 |
| | HydraLoRA ($r$=8) | 11.10M | 87.34 | 92.67 | 90.16 | 58.83 | 27.95 | 75.50 | 72.08 |
| | TopLoRA ($r$=8) | 6.88M | 87.85 | 94.78 | 91.80 | 58.43 | 30.97 | 74.87 | 73.11 |
| | TopLoRA ($r$=16) | 13.80M | 86.33 | 94.83 | 92.52 | 59.29 | 31.10 | 75.60 | 73.28 |
| | **Ours** ($r$=8) | 6.89M | 88.35 | **97.33** | 94.69 | 65.28 | 35.04 | **78.30** | 76.50 |
| | **Ours** ($r$=16) | 13.80M | **90.13** | 96.50 | **96.06** | **66.03** | **36.61** | 77.10 | **77.07** |
| LLaMA-3-8B | LoRA ($r$=8) | 4.72M | 82.28 | 87.06 | 91.60 | 55.65 | 24.02 | 68.53 | 68.19 |
| | LoRA ($r$=16) | 9.44M | 84.56 | 91.22 | 92.26 | 57.22 | 25.72 | 70.17 | 70.19 |
| | LoRA ($r$=32) | 18.90M | 87.17 | 93.39 | 93.50 | 57.87 | 26.25 | 71.83 | 71.67 |
| | DoRA ($r$=16) | 9.63M | 85.95 | 89.67 | 92.62 | 56.52 | 26.19 | 70.40 | 70.23 |
| | MELoRA ($r$=16) | 9.44M | 85.82 | 87.83 | 91.54 | 55.34 | 24.41 | 71.20 | 69.36 |
| | HydraLoRA ($r$=8) | 9.04M | 86.08 | 91.00 | 91.14 | 55.50 | 25.98 | 68.10 | 69.63 |
| | TopLoRA ($r$=8) | 7.87M | 87.34 | 92.83 | 92.91 | 59.21 | 24.02 | 74.70 | 71.84 |
| | TopLoRA ($r$=16) | 15.70M | 88.86 | 92.17 | 93.31 | 61.11 | 28.74 | 73.50 | 72.95 |
| | **Ours** ($r$=8) | 7.88M | 87.85 | **94.17** | **96.46** | 65.13 | 35.43 | **77.10** | 76.02 |
| | **Ours** ($r$=16) | 15.80M | **91.14** | 93.67 | 96.06 | 64.29 | **38.19** | 76.00 | **76.55** |
| Qwen3-8B | LoRA ($r$=8) | 5.31M | 90.38 | 96.83 | 93.70 | 74.00 | 30.71 | 85.50 | 78.52 |
| | LoRA ($r$=16) | 10.62M | 90.63 | 96.50 | 92.91 | 75.89 | 35.04 | 84.40 | 79.23 |
| | LoRA ($r$=32) | 21.23M | 91.14 | 96.67 | 93.90 | 75.21 | 36.61 | 86.20 | 79.95 |
| | DoRA ($r$=16) | 10.84M | 90.13 | 94.83 | 92.91 | 77.33 | 40.16 | 83.40 | 79.79 |
| | MELoRA ($r$=16) | 10.62M | 90.89 | 95.00 | 91.73 | 76.04 | 37.01 | 87.30 | 79.66 |
| | HydraLoRA ($r$=8) | 10.17M | **93.67** | 96.67 | 92.72 | 75.66 | 30.71 | 86.90 | 79.39 |
| | TopLoRA ($r$=8) | 8.85M | 89.37 | 97.83 | 93.50 | 77.10 | **43.31** | 83.40 | 80.75 |
| | TopLoRA ($r$=16) | 17.70M | 87.59 | 98.83 | 96.85 | 83.62 | 38.98 | 79.70 | 80.93 |
| | **Ours** ($r$=8) | 8.86M | 91.39 | 98.50 | 97.64 | **85.29** | **43.31** | **87.70** | 83.97 |
| | **Ours** ($r$=16) | 17.76M | 93.16 | **99.17** | **97.83** | 84.69 | 42.91 | 87.60 | **84.23** |
| Qwen2.5-14B | LoRA ($r$=8) | 8.65M | **93.16** | 96.67 | 92.32 | 75.66 | 31.10 | 85.60 | 79.09 |
| | LoRA ($r$=16) | 17.30M | 91.90 | 96.33 | 92.91 | 74.37 | 34.65 | 86.40 | 79.43 |
| | LoRA ($r$=32) | 34.60M | 92.24 | 97.39 | 92.98 | 76.37 | 34.78 | 87.13 | 80.15 |
| | DoRA ($r$=16) | 17.60M | 92.41 | 96.39 | 92.39 | 75.92 | 36.22 | 86.80 | 80.02 |
| | MELoRA ($r$=16) | 17.30M | 92.91 | 97.33 | 92.13 | 75.89 | 33.86 | 85.60 | 79.62 |
| | HydraLoRA ($r$=8) | 16.40M | 92.41 | 96.22 | 92.45 | 76.32 | 36.61 | 86.97 | 80.16 |
| | TopLoRA ($r$=8) | 14.60M | 91.31 | 97.67 | 93.44 | 77.31 | 35.96 | **87.43** | 80.52 |
| | TopLoRA ($r$=16) | 29.10M | 91.65 | 98.50 | 93.90 | 75.74 | 37.40 | 87.40 | 80.76 |
| | **Ours** ($r$=8) | 14.60M | 90.63 | 98.50 | **97.44** | **86.13** | 44.09 | 86.40 | 83.86 |
| | **Ours** ($r$=16) | 29.20M | 92.15 | **99.17** | **97.44** | **86.13** | **45.28** | 86.10 | **84.38** |

decomposes pretrained weights into magnitude and direction components and applies low-rank updates only to the directional part, (3) **MELoRA** (Ren et al., 2024) improves expressiveness by structurally combining multiple low-rank adapters, (4) **HydraLoRA** (Tian et al., 2024) employs multi-branch low-rank structures with routing mechanisms, and (5) **TopLoRA** (Li et al., 2025) introduces token-dependent modulation on low-rank updates to enable finer-grained adaptation. These methods represent different lines of improvement over LoRA, including reparameterization, structured low-rank designs, and input-conditioned mechanisms. Implementation and configuration details of all baselines (including rank, number of branches or groups, and parameter budget alignment) are summarized in Appendix B.

## 4.2. Results on Mathematical Reasoning Tasks

Table 1 shows that U-LoRA achieves the best average accuracy across all four decoder-only LLMs, consistently outperforming vanilla LoRA and representative recent variants under comparable parameter budgets. On LLaMA-3-8B, U-LoRA reaches 76.02% with $r$=8, exceeding LoRA with a much larger rank ($r$=32) at 71.67%, and further improves to 76.55% with $r$=16. This trend suggests that improving the utilization of low-rank directions is more effective than simply increasing the rank of LoRA. The same pattern holds on the stronger Qwen2.5-14B backbone, where U-LoRA attains 83.86% and 84.38% with $r$=8 and $r$=16, respectively, surpassing both LoRA and recent variants under similar parameter budgets. Notably, U-LoRA also yields larger

*Table 2.* GLUE benchmark results of different parameter-efficient fine-tuning methods on RoBERTa-Base and RoBERTa-Large models. The highest average precision is bolded, and the second-highest is underlined.

| Model | Method | #Params | RTE | MRPC | STS-B | CoLA | SST-2 | QNLI | MNLI | QQP | Avg |
|---|---|---|---|---|---|---|---|---|---|---|---|
| RoBERTa-Base | LoRA ($r$=8) | 0.29M | 72.56 | 87.25 | 87.12 | 56.10 | 93.46 | 91.58 | 84.89 | 87.46 | 82.55 |
| | LoRA ($r$=16) | 0.59M | 72.92 | 87.99 | 87.46 | 55.21 | 93.81 | 91.89 | 85.52 | 87.79 | 82.82 |
| | LoRA ($r$=32) | 1.18M | 75.09 | 89.22 | 88.01 | 58.58 | 93.58 | 90.12 | 85.84 | 88.37 | 83.60 |
| | DoRA ($r$=16) | 0.61M | 74.85 | 87.83 | 88.48 | 56.46 | 93.39 | 91.86 | 85.25 | 87.89 | 83.25 |
| | MELoRA ($r$=16) | 0.59M | 75.45 | 88.73 | 87.27 | 54.43 | 93.00 | 91.51 | 84.93 | 87.53 | 82.86 |
| | HydraLoRA ($r$=8) | 0.65M | 73.65 | 89.46 | 88.53 | 57.03 | 93.23 | 91.89 | 85.52 | 87.57 | 83.36 |
| | TopLoRA ($r$=8) | 0.44M | **78.34** | 87.99 | 90.05 | 58.55 | 92.43 | 92.02 | 85.60 | 88.12 | 84.14 |
| | TopLoRA ($r$=16) | 0.89M | **78.34** | 88.73 | 88.90 | 60.34 | 93.35 | 92.11 | 85.65 | 88.43 | 84.48 |
| | **Ours** ($r$=8) | 0.45M | 77.26 | **90.20** | **90.62** | 60.61 | **94.04** | **92.22** | 85.72 | 88.37 | 84.88 |
| | **Ours** ($r$=16) | 0.90M | 77.26 | 89.95 | 90.50 | **61.57** | 93.92 | 92.20 | **86.13** | **88.48** | **85.00** |
| RoBERTa-Large | LoRA ($r$=8) | 0.79M | 71.96 | 88.40 | 89.88 | 59.76 | 95.41 | 93.07 | 88.67 | 87.89 | 84.38 |
| | LoRA ($r$=16) | 1.57M | 77.74 | 88.48 | 90.60 | 61.23 | 95.57 | 93.72 | 89.29 | 88.25 | 85.61 |
| | LoRA ($r$=32) | 3.15M | 81.35 | 89.64 | 91.45 | 60.90 | 95.60 | 93.76 | 89.52 | 88.61 | 86.35 |
| | DoRA ($r$=16) | 1.62M | 80.14 | 88.73 | 90.94 | 61.73 | 95.68 | 93.45 | 89.30 | 88.29 | 86.03 |
| | MELoRA ($r$=16) | 1.57M | 79.48 | 87.75 | 90.17 | 60.59 | 95.87 | 93.21 | 88.86 | 87.89 | 85.48 |
| | HydraLoRA ($r$=8) | 1.72M | 79.42 | 89.46 | 90.63 | 61.07 | 95.76 | 93.39 | 89.25 | 88.26 | 85.91 |
| | TopLoRA ($r$=8) | 1.18M | 80.51 | 89.30 | 91.54 | 61.75 | 95.64 | 93.89 | 89.66 | 88.83 | 86.39 |
| | TopLoRA ($r$=16) | 2.36M | 85.20 | 90.44 | 91.50 | 64.56 | 95.64 | 94.20 | 89.94 | 88.94 | 87.55 |
| | **Ours** ($r$=8) | 1.19M | 85.92 | 90.93 | 91.68 | 66.51 | **95.99** | **94.67** | **90.15** | 89.17 | 88.13 |
| | **Ours** ($r$=16) | 2.39M | **86.28** | **91.18** | **91.99** | **66.79** | 95.76 | 94.65 | **90.15** | **89.49** | **88.29** |

gains on more challenging reasoning tasks. For example, on Qwen3-8B, U-LoRA improves performance on AQuA to 43.31 (vs. 36.61 for LoRA with $r$=32) and on GSM8K to 85.29 (vs. 75.21 for LoRA with $r$=32), indicating that its advantage is more pronounced in settings requiring multi-step reasoning and discrete choice-based inference.

### 4.3. Results on Natural Language Understanding Tasks

As shown in Table 2, U-LoRA achieves the best average performance on both RoBERTa-Base and RoBERTa-Large, consistently outperforming vanilla LoRA and recent variants under comparable parameter budgets. In RoBERTa-Base, U-LoRA reaches 84.88%/85.00% with $r$=8/16, surpassing TopLoRA with $r$=16 (84.48%) and LoRA with $r$=32 (83.60%). In RoBERTa-Large, it achieves 88.13%/88.29% with $r$=8/16, maintaining the same advantage. In contrast, LoRA with natively increasing ranks achieves only marginal gains. For instance, moving from $r$=16 to $r$=32 improves the average score only from 82.82% to 83.60% on RoBERTa-Base and from 85.61% to 86.35% on RoBERTa-Large. In particular, even with a lower rank (e.g. $r$=8), U-LoRA is competitive with higher-rank LoRA and more complex baselines, suggesting that the improvements are not driven primarily by parameter scaling, but by more effective use of a fixed low-rank budget. From the individual-task perspective, U-LoRA achieves larger gains in CoLA (linguistic acceptability) on the two foundational models (RoBERTa-Base: 60.61% vs 58.58% and RoBERTa-Large: 66.51% vs 60.90%, compared to LoRA $r$=32), indicating particularly advantageous for tasks requiring fine-grained syntactic modeling. U-LoRA also benefits from stable representation alignment in STS-B (RoBERTa-Base: 90.62% vs 90.05% and RoBERTa-Large: 91.68% vs 91.54%, compared to TopLoRA $r$=8), and remains competitive on smaller datasets such as RTE, supporting consistent improvements across tasks and model scales.

## 5. Discussion

### 5.1. Ablation Study

*Table 3.* Component ablations of U-LoRA on Qwen3-8B for mathematical reasoning. We report the unweighted mean accuracy (AVG) across benchmarks; $\Delta$ denotes the change relative to FULL. Components: token-wise utilization (TOK), sequence-level aggregation (SEQ), and EMA historical prior (EMA). Details of all results are provided in Appendix C.

| Method | AVG | $\Delta$ |
|---|---|---|
| U-LoRA (Full) | **83.97** | **+0.00** |
| w/o TOK | 78.95 | -5.02 |
| w/o SEQ | 79.16 | -4.81 |
| w/o EMA | 79.66 | -4.31 |
| w/o TOK, SEQ | 78.90 | -5.07 |
| w/o TOK, EMA | 80.32 | -3.65 |
| w/o SEQ, EMA | 79.24 | -4.73 |
| w/o TOK, SEQ, EMA (LoRA) | 78.52 | -5.45 |

As shown in Table 3, we investigate the effectiveness of each component in U-LoRA. The results demonstrate that all modules contribute to the final gains, as independent removal of each component consistently degrades perfor-

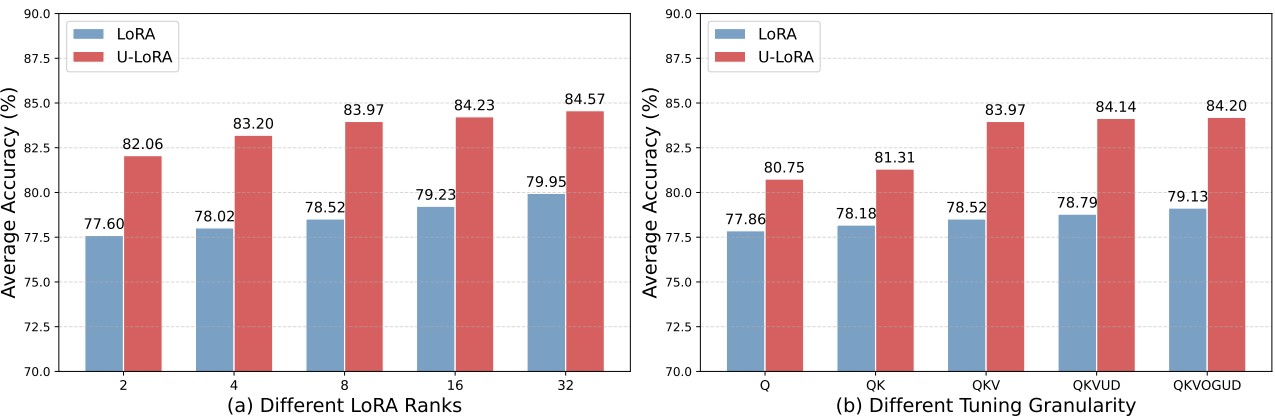

*Figure 3.* **Scalability analysis on mathematical reasoning tasks using Qwen3-8B.** (a) Comparison across different LoRA ranks. (b) Impact of target modules (tuning granularity) with rank $r=8$.

mance. In particular, TOK has the largest impact, with the average performance dropping from 83.97 to 78.95 (a decrease of 5.02 points), while SEQ exhibits a similar trend, with performance decreasing from 83.97 to 79.16 (a reduction of 4.81 points). Furthermore, Appendix C shows the results for each benchmark in details. SEQ leads to particularly severe performance degradation on AQuA, suggesting that sequence-level modeling is crucial for algebraic multiple-choice reasoning. In addition, multiple components jointly cooperation results in further performance drops, highlighting the complementary effects among components. Specifically, the removal of all components reduces the average score to 78.52 (a decrease of 5.45 points), which matches the corresponding LoRA baseline.

## 5.2. Scalability Analysis

We evaluate the scalability of U-LoRA with respect to adaptation capacity, tuning granularity, and model scale on mathematical reasoning tasks, including instruct-tuned variants. Figure 3 reports *average* accuracy, with detailed per-task results deferred to Appendix C.

**Effect of LoRA rank.** We sweep the LoRA rank over $\{2, 4, 8, 16, 32\}$. As shown in Figure 3(a), U-LoRA consistently outperforms LoRA across all tested ranks, including the low-rank regime, suggesting that the gains primarily come from better utilization of the existing low-rank subspace rather than expanding its dimensionality.

**Effect of tuning granularity.** We further vary the target modules beyond the default attention projections. Specifically, we consider **Q**, **QK**, **QKV**, **QKVUD**, and **QKVOGUD**, where O/G/U/D denote the output, gate, up, and down projections, respectively. Under a fixed rank $r=8$, Figure 3(b) shows that U-LoRA remains consistently better than LoRA across different tuning granularities, indicating

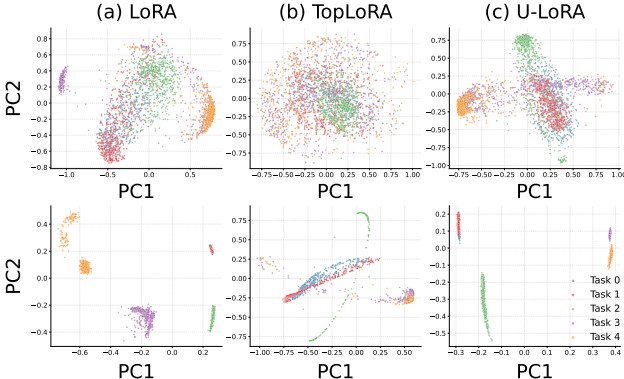

*Figure 4.* PCA visualization of low-rank parameter updates $\Delta Q$ for RoBERTa-Large during multi-task fine-tuning. Implementation details are provided in the Appendix B.

robustness to module choices.

**Effect of model scale and instruct tuning.** We additionally scale up model size to 72B and evaluate on instruct-tuned variants. Our results show that U-LoRA remains effective at the 72B scale, consistent with trends observed on 8B and 14B models. On instruct models, U-LoRA continues to yield significant improvements (e.g., +3.24 over TopLoRA on 14B-Instruct), indicating that it is also effective for models that already possess strong task capabilities. Detailed results for these scaling and instruct-tuning experiments are provided in Appendix C.

## 5.3. Geometry of Low-Rank Parameter Updates

Figure 4 visualizes the geometric structure of low-rank parameter updates at different depths. We analyze both intra-task consistency (cluster compactness) and inter-task structure (semantic relationships across tasks). Across all methods, updates at shallow layers exhibit more dispersed

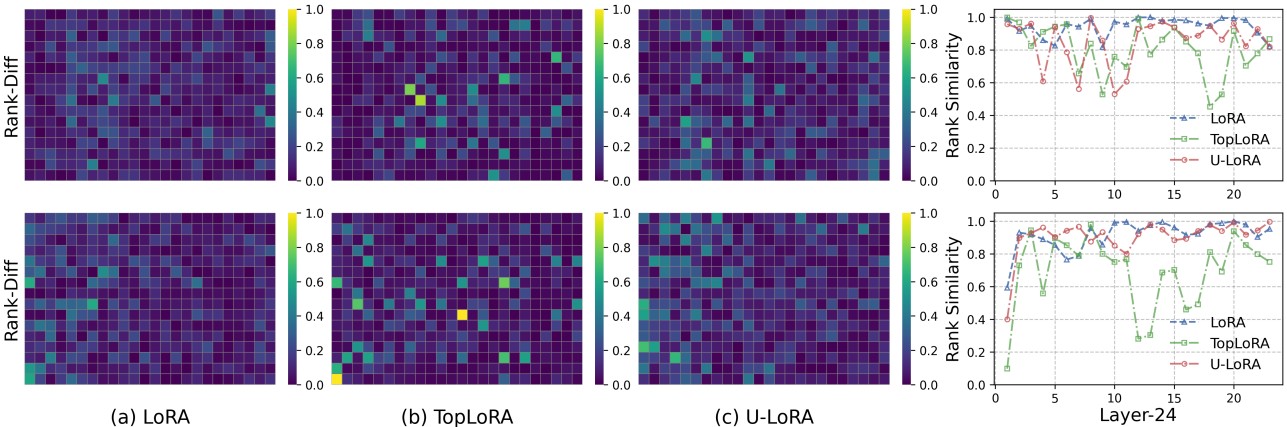

*Figure 5.* Depth-wise differences in low-rank utilization on RoBERTa-Large. Heatmaps report $\left|\hat{v}_1^{(l)} - \hat{v}_2^{(l)}\right|$ for $\hat{v}^{(l)} = v^{(l)}/\|v^{(l)}\|_2$ (rank representation before $B$). **Left:** layer–rank difference heatmaps for two controlled comparisons: (**Top**) the same token `book` under verb vs. noun contexts; (**Bottom**) different tokens (`a` vs. `book`) within the same sentence. **Right:** the corresponding rank-wise cosine similarity across layers.

patterns, reflecting sensitivity to local token-level variations. As depth increases, update distributions become more structured, indicating a transition from token-specific adjustments toward task-level representations in deeper layers. TopLoRA produces relatively scattered representations across both shallow and deep layers, without clear clustering structure. This suggests that token-wise adaptive modulation introduces high directional diversity in parameter updates but fails to preserve task identity, resulting in poor task separability and weak semantic organization. LoRA exhibits partial clustering in deeper layers; however, certain tasks (e.g., Task 3 and Task 4) still show weak intra-task compactness, and inter-task relationships remain relatively ambiguous, indicating limited task discrimination in the update space. Notably, U-LoRA forms the most compact and well-structured clusters in deeper layers, with updates from the same task showing stronger alignment, and semantically related tasks (e.g., Task 0 and Task 1, Task 3 and Task 4) exhibiting closer proximity in the update space. By incorporating sequence modeling and historical priors, U-LoRA maintains both local task coherence and global semantic structure in the learned representation space.

### 5.4. Case Study

We use a case study to illustrate how different adapters utilize a fixed low-rank subspace across depth. For context variation, we consider the polysemous token `book` in "I can book a flight." (verb) versus "This is a good book." (noun). We also compare `a` and `book` within the same sentence to probe token-role differences. Figure 5 reports both layer–rank differences and layer-wise cosine similarity for these comparisons.

**Same token, different contexts.** As shown in Figure 5(top), when comparing the same token `book` under two different contexts, LoRA maintains relatively higher similarity across most layers, overall higher than TopLoRA and U-LoRA. This indicates that under shared and input-agnostic low-rank updates, the low-rank utilization patterns of the same token remain more similar across contexts, making context-dependent differences less explicit in the low-rank subspace. In contrast, both TopLoRA and U-LoRA exhibit lower similarity in the middle and upper layers, suggesting that introducing token-level modulation increases the context sensitivity of low-rank utilization. However, their layer-wise behaviors differ noticeably: TopLoRA shows no consistent trend across layers, reflecting greater layer-wise instability, whereas U-LoRA follows a smoother trajectory and gradually stabilizes in later layers. This contrast suggests that relying solely on token-level signals may lead to uncoordinated utilization decisions across layers within a sentence, while incorporating sequence-level information helps form more coherent, context-dependent low-rank utilization patterns as representations become more abstract.

**Different tokens, same sentence.** As shown in Figure 5(bottom), when comparing the function word `a` and the content word `book` within the same sentence, all three methods exhibit stronger differences in the lower layers. This observation is consistent with prior findings that shallow layers tend to emphasize lexical or syntactic distinctions between tokens. As depth increases, the similarity under LoRA rises rapidly, indicating that deeper representations gradually shift toward more abstract, sequence-level integration. In contrast, TopLoRA maintains noticeable layer-wise fluctuations in similarity in the middle and upper layers, suggesting that when relying solely on token-level modula-

*Table 4.* Efficiency–Performance Trade-off on Mathematical Reasoning Tasks (Qwen3-8B, $r = 8$).

| Method | Avg. Acc. (%) | Gain | Task Lat. ($\times$) | Extra Lat. | Gain / $0.1\times$ | Output Tok. ($\times$) | Token Lat. ($\times$) |
|---|---|---|---|---|---|---|---|
| LoRA | 78.52 | — | $1.00\times$ | — | — | $1.00\times$ | $1.00\times$ |
| TopLoRA | 80.75 | +2.23 | $1.28\times$ | +0.28$\times$ | 0.8 | $1.02\times$ | $1.25\times$ |
| U-LoRA | 83.97 | +5.45 | $1.39\times$ | +0.39$\times$ | 1.4 | $1.07\times$ | $1.30\times$ |

tion, token-specific low-rank utilization differences are less likely to naturally converge into stable, abstract representations in deeper layers. By comparison, U-LoRA exhibits a clearer "separate-then-integrate" pattern, preserving token-level distinctions in intermediate layers while gradually returning to higher similarity in deeper layers. This behavior is consistent with the goal of introducing sequence-level coordination on top of token-conditioned modulation.

### 5.5. Latency discussion

We conduct a detailed latency analysis to identify the sources of runtime overhead, with the full quantitative results provided in Appendix D. As shown in Table 4, our results indicate that for every $0.1\times$ increase in latency, U-LoRA achieves a performance gain of +1.40, which is $1.75\times$ higher than TopLoRA (+0.80), demonstrating a better efficiency–performance trade-off. Additionally, we report the average output token length for math reasoning tasks. Under this finer-grained analysis, part of the increased latency can be attributed to longer generated sequences. More fine-grained low-rank adaptation improves answer quality, which in turn leads to longer reasoning chains. When accounting for this factor, the observed latency of both U-LoRA and TopLoRA decreases. Overall, U-LoRA achieves a favorable efficiency–performance trade-off. Although dynamic adaptation introduces additional system-level overhead, the resulting improvements in solution quality suggest that this overhead is justified.

### 6. Conclusion

In this paper, we study LoRA from the perspective of low-rank subspace utilization and identify that a key limitation comes from the implicit and largely uniform use of low-rank directions across inputs and training stages. Motivated by this view, we proposed U-LoRA, which preserves LoRA's low-rank parameterization while learning sequence-conditioned, token-wise utilization coefficients and stabilizing them with a bias-corrected historical prior. Experiments across diverse backbones and benchmarks show consistent improvements over representative LoRA baselines and variants under comparable parameter budgets. These results highlight utilization-aware modeling as an effective direction for parameter-efficient fine-tuning.

### Acknowledgements

This work is supported by the Fundamental Research Funds for the Central Universities of China under Grant 2024JBGP008 and the National Natural Science Foundation of China (No. 62406018, 62476023, 61976016, 62376019), and the authors would like to thank the anonymous reviewers for their valuable comments and suggestions to improve this paper.

### Impact Statement

This paper presents work whose goal is to advance the field of Machine Learning. There are many potential societal consequences of our work, none which we feel must be specifically highlighted here.

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

## A. Dataset Details

**Mathematical Reasoning Tasks.** We fine-tune models on Math10K (Hu et al., 2023) and evaluate on six arithmetic reasoning benchmarks. Math10K is mainly sourced from GSM8K (Cobbe et al., 2021) and AQuA (Ling et al., 2017), and covers diverse arithmetic problem formulations. Our evaluation suite includes MultiArith (Roy & Roth, 2015) and AddSub (Hosseini et al., 2014) (multi-step arithmetic word problems with numeric answers), GSM8K (grade-school math word problems requiring multi-step reasoning), AQuA (multiple-choice algebraic word problems), SingleEq (Koncel-Kedziorski et al., 2015) (single-equation word problems), and SVAMP (Patel et al., 2021) (robustness under varied problem structures). Dataset statistics are summarized in Table 5.

*Table 5.* Statistics of the arithmetic reasoning benchmark datasets.

| Dataset | Domain | Train | Test | Answer Type |
|---------|--------|-------|------|-------------|
| MultiArith | Math | – | 600 | Number |
| AddSub | Math | – | 395 | Number |
| GSM8K | Math | 8.8k | 1,319 | Number |
| AQuA | Math | 100k | 254 | Option |
| SingleEq | Math | – | 508 | Number |
| SVAMP | Math | – | 1,000 | Number |

**Natural Language Understanding Tasks.** We evaluate on the GLUE benchmark (Wang et al., 2018), which covers textual entailment (RTE (Bentivogli et al., 2009), QNLI (Rajpurkar et al., 2016), MNLI (Williams et al., 2018)), paraphrase identification (MRPC (Dolan & Brockett, 2005), QQP (Wang et al., 2018)), semantic textual similarity (STS-B), linguistic acceptability judgment (CoLA (Warstadt et al., 2019)), and sentiment analysis (SST-2 (Socher et al., 2013)). Dataset statistics are summarized in Table 6.

*Table 6.* Statistics of the GLUE benchmark datasets.

| Corpus | Train | Valid | Test | Metric |
|--------|-------|-------|------|--------|
| RTE | 2.5k | 277 | 3k | Accuracy |
| MRPC | 3.7k | 408 | 1.7k | Accuracy |
| STS-B | 5.7k | 1.5k | 1.4k | Pearson corr. |
| CoLA | 8.5k | 1,043 | 1,063 | Matthews corr. |
| SST-2 | 67k | 872 | 1.8k | Accuracy |
| QNLI | 105k | 5.5k | 5.5k | Accuracy |
| QQP | 364k | 40.4k | 391k | Accuracy |
| MNLI | 393k | 20k | 20k | Accuracy |

## B. Experimental Settings

**Implementation.** All experiments are implemented in PyTorch with models and datasets from Hugging Face. We run experiments on NVIDIA GeForce RTX 3090 GPUs.

**Mathematical Reasoning Tasks.** For baselines, we set DoRA rank to 16. HydraLoRA uses rank 8 with three $B$ matrices, and MELoRA uses Mini-LoRA rank 16 with four groups. We use AdamW with $\beta_1$=0.9, $\beta_2$=0.999, $\varepsilon$=$10^{-8}$, and no weight decay. We select the learning rate from $\{10^{-4}, 2 \times 10^{-4}, 3 \times 10^{-4}, 10^{-3}\}$ and is set to $10^{-4}$.

For LoRA-based methods, we set the scaling parameter as $\alpha = 2r$, resulting in a constant scaling factor $\gamma = \alpha/r = 2$. This setting is consistently applied to all LoRA baselines with ranks $r \in \{8, 16, 32\}$, following the default configuration adopted in TopLoRA (Li et al., 2025). Unless otherwise stated, we do not further tune the scaling strategy for individual baselines. LoRA modules are applied to the query, key, and value projection weights with a dropout rate of 0.05, using BF16 precision, following the same configuration as TopLoRA. We use a warmup of 100 steps, batch size 16, and maximum sequence length 256. Training is conducted for one epoch. In all experiments, we set the momentum parameter $\beta$ to 0.99, which provides a stable and robust trade-off across tasks. A detailed analysis is reported in Section C. Four models are employed: Gemma-7B (Team et al., 2024), LLaMA-3-8B (Dubey et al., 2024), Qwen3-8B-Base (Yang et al., 2025) and

*Table 7.* Component ablations of U-LoRA on Qwen3-8B for mathematical reasoning. We report accuracies on each benchmark; Avg denotes the unweighted mean. Components: (1) token-wise utilization, (2) sequence-level aggregation, and (3) EMA historical prior.

| Method | AddSub | MultiArith | SingleEq | GSM8K | AQuA | SVAMP | Avg |
|---|---|---|---|---|---|---|---|
| U-LoRA (Full) | 91.39 | 98.50 | 97.64 | 85.29 | 43.31 | 87.70 | 83.97 |
| w/o TOK | 89.87 | 91.67 | 90.94 | 76.57 | 40.55 | 84.10 | 78.95 |
| w/o SEQ | 90.13 | 97.67 | 94.29 | 74.60 | 31.89 | 86.40 | 79.16 |
| w/o EMA | 90.89 | 95.00 | 91.73 | 76.04 | 37.01 | 87.30 | 79.66 |
| w/o TOK, SEQ | 91.90 | 97.00 | 92.72 | 74.37 | 31.50 | 85.90 | 78.90 |
| w/o TOK, EMA | 93.16 | 90.67 | 91.93 | 75.66 | 42.91 | 87.60 | 80.32 |
| w/o SEQ, EMA | 89.87 | 97.50 | 94.09 | 75.13 | 33.46 | 85.40 | 79.24 |
| w/o TOK, SEQ, EMA (LoRA) | 90.38 | 96.83 | 93.70 | 74.00 | 30.71 | 85.50 | 78.52 |

Qwen2.5-14B (Yang et al., 2024). In this work, we use Qwen3-8B as a shorthand notation for Qwen3-8B-Base, unless explicitly stated otherwise.

**Natural Language Understanding Tasks.** We train for 10 epochs on RTE and MRPC, 5 epochs on STS-B and CoLA, 2 epochs on SST-2 and QNLI, and 1 epoch on MNLI and QQP. We use learning rate $3 \times 10^{-4}$ for RoBERTa-Base and $10^{-4}$ for RoBERTa-Large, with warmup ratio 0.03. Batch size is 32 and maximum sequence length is 512. All methods are applied to the query and value weights, following standard practice in prior work. For U-LoRA, the momentum parameter is fixed to $\beta = 0.99$ across all experiments.

**The PCA Visualization** To analyze the impact of historical priors on the weight representation space, we conducted visualization experiments using a subset of the GLUE benchmark set. Specifically, we analyze the parameter update trajectories on a subset of the GLUE benchmark: {MRPC, QNLI, QQP, SST-2, RTE}, denoted as Task 0 to Task 4, respectively. We perform one epoch of multi-task training using the RoBERTa-Large backbone. During each forward pass, we record the incremental update matrix $\Delta W_Q$ of the Query projection. For each sampled token $x_t$, we compute its induced update representation $\Delta q_t = x_t \cdot \Delta W_Q \in \mathbb{R}^{d_{\mathrm{model}}}$. We randomly sample $N = 500$ tokens per task and extract their corresponding $\Delta q_t$ vectors from a shallow layer (Layer 2) and a deep layer (Layer 22). We then apply Principal Component Analysis (PCA) to project these $d_{\mathrm{model}}$-dimensional representations into a two-dimensional space. This visualization allows us to inspect the directional consistency of updates across tokens and the geometric alignment between different tasks under the influence of historical priors.

# C. Additional Experimental Results

**Detailed Results of the Ablation Study** In Section 5.1, we conduct a comprehensive ablation study and analyze the impact of individual components in U-LoRA. For clarity, we present only the averaged results in Table 3, while the complete benchmark-wise results are reported in Table 7 for reference. The ablation results are evaluated on Qwen3-8B for mathematical reasoning tasks, where the contributions of token-wise utilization, sequence-level aggregation, and the EMA-based historical prior are examined in isolation and in combination.

**Scalability Analysis** In Section 5.2, we evaluate the scalability of U-LoRA on mathematical reasoning tasks using Qwen3-8B from two complementary perspectives: the LoRA rank and the tuning granularity. Figure 3 reports the average accuracy under different configurations, averaged over sub-tasks, while the complete results are provided in Tables 8 and 9 for detailed comparison. In addition, we analyze the training dynamics of U-LoRA. As shown in Table 10, U-LoRA exhibits lower training loss from the early stages and consistently outperforms all baselines throughout the optimization process. This indicates that it enables more sufficient and stable learning across different types of samples. The convergence advantage is consistently observed across all baselines, demonstrating that the adaptive utilization mechanism not only improves final performance but also accelerates optimization. Figure 6 presents the training loss curves of LoRA and U-LoRA under four model architectures. Across all settings, U-LoRA consistently exhibits faster loss convergence than standard LoRA, suggesting more efficient optimization and improved adaptation efficiency during fine-tuning. Furthermore, we scale U-LoRA to 72B parameters and evaluate on instruction-tuned variants of Qwen2.5. As shown in Table 11, U-LoRA

effectively transfers to larger scales and instruction-tuned models.

*Table 8.* The accuracy of LoRA and U-LoRA with varying target modules on mathematical reasoning tasks using Qwen3-8B.

| Target | Method | #Params | AddSub | MultiArith | SingleEq | GSM8K | AQuA | SVAMP | Avg |
|---|---|---|---|---|---|---|---|---|---|
| Q | LoRA | 2.36M | 89.87 | 96.33 | 92.13 | 73.92 | 31.89 | 83.00 | 77.86 |
| | U-LoRA | 3.54M | 89.37 | 97.83 | 93.50 | 77.10 | 43.31 | 83.40 | 80.75 |
| QK | LoRA | 3.83M | 86.84 | 97.67 | 90.94 | 75.89 | 36.22 | 81.50 | 78.18 |
| | U-LoRA | 6.20M | 87.85 | 98.83 | 97.05 | 83.85 | 40.16 | 80.10 | 81.31 |
| QKV | LoRA | 5.31M | 90.38 | 96.83 | 93.70 | 74.00 | 30.71 | 85.50 | 78.52 |
| | U-LoRA | 8.86M | 91.39 | 98.50 | 97.64 | 85.29 | 43.31 | 87.70 | 83.97 |
| QKVUD | LoRA | 14.75M | 90.38 | 96.17 | 93.50 | 75.13 | 33.07 | 84.50 | 78.79 |
| | U-LoRA | 23.03M | 93.67 | 99.00 | 96.85 | 84.08 | 40.55 | 90.70 | 84.14 |
| QKVOGUD | LoRA | 21.82M | 91.39 | 96.67 | 94.29 | 75.06 | 32.28 | 85.10 | 79.13 |
| | U-LoRA | 32.48M | 93.16 | 99.33 | 97.05 | 85.75 | 43.70 | 86.20 | 84.20 |

*Table 9.* The accuracy of LoRA and U-LoRA with varying ranks on mathematical reasoning tasks using Qwen3-8B.

| Rank | Method | #Params | AddSub | MultiArith | SingleEq | GSM8K | AQuA | SVAMP | Avg |
|---|---|---|---|---|---|---|---|---|---|
| $r = 2$ | LoRA | 1.33M | 90.13 | 95.83 | 92.91 | 74.45 | 28.35 | 83.90 | 77.60 |
| | U-LoRA | 2.21M | 90.38 | 98.67 | 93.31 | 79.83 | 44.49 | 85.70 | 82.06 |
| $r = 4$ | LoRA | 2.65M | 87.34 | 97.83 | 92.91 | 73.92 | 31.10 | 85.00 | 78.02 |
| | U-LoRA | 4.43M | 93.42 | 99.00 | 96.06 | 84.23 | 38.98 | 87.50 | 83.20 |
| $r = 8$ | LoRA | 5.31M | 90.38 | 96.83 | 93.70 | 74.00 | 30.71 | 85.50 | 78.52 |
| | U-LoRA | 8.86M | 91.39 | 98.50 | 97.64 | 85.29 | 43.31 | 87.70 | 83.97 |
| $r = 16$ | LoRA | 10.62M | 90.63 | 96.50 | 92.91 | 75.89 | 35.04 | 84.40 | 79.23 |
| | U-LoRA | 17.76M | 93.16 | 99.17 | 97.83 | 84.69 | 42.91 | 87.60 | 84.23 |
| $r = 32$ | LoRA | 21.23M | 91.14 | 96.67 | 93.90 | 75.21 | 36.61 | 86.20 | 79.95 |
| | U-LoRA | 35.62M | 93.67 | 99.17 | 96.46 | 84.31 | 43.70 | 90.10 | 84.57 |

**Effect of the momentum parameter $\beta$.** We further investigate the impact of the hyperparameter $\beta$ in U-LoRA on a range of mathematical reasoning benchmarks using Qwen3-8B, as summarized in Table 12. Overall, U-LoRA exhibits moderate sensitivity to the choice of $\beta$, moderate-to-large $\beta$ improves stability and yields better average performance, with the best result at $\beta = 0.995$. Further increasing $\beta$ slightly degrades the average score (e.g., $\beta \geq 0.998$), suggesting that excessive inertia may hinder timely adaptation to task-specific signals. Considering both performance robustness and experimental reproducibility, we adopt $\beta = 0.99$ as the default setting in all experiments. Although this choice is not strictly optimal, it provides a strong and stable trade-off across tasks without requiring additional hyperparameter tuning.

**Standard Deviation Analysis** In the main experiments, each setting was repeated three times, and the average results are reported. For conciseness, standard deviations are provided in the Appendix. Table 13 shows the standard deviations for each GLUE dataset, where a separate model was trained for each. Table 14 presents the standard deviation of the average accuracy for the mathematical reasoning tasks, where a single model was used across these sub-tasks. Notably, the standard deviation remains stable and is much smaller than the accuracy improvement achieved by U-LoRA.

## D. Limitations

A primary limitation of U-LoRA lies in inference efficiency. As shown in Table 4, U-LoRA exhibits a moderate increase in inference latency compared to standard LoRA. This is primarily due to its token-wise adaptive LoRA weights, which cannot be statically merged into the pretrained parameters after fine-tuning and therefore require additional computations at inference time.

*Table 10.* Training Loss and Accuracy Comparison Across Methods (Qwen3-8B).

| Method | 50 Steps | 100 Steps | 200 Steps | 300 Steps | 400 Steps | Final Loss | Avg |
|--------|----------|-----------|-----------|-----------|-----------|------------|-----|
| LoRA | 0.3657 | 0.2759 | 0.2788 | 0.2711 | 0.2615 | 0.2710 | 78.52 |
| DoRA | 0.3549 | 0.2713 | 0.2747 | 0.2662 | 0.2559 | 0.2652 | 79.79 |
| TopLoRA | 0.3494 | 0.2707 | 0.2746 | 0.2638 | 0.2505 | 0.2572 | 80.75 |
| U-LoRA | 0.3418 | 0.2694 | 0.2745 | 0.2592 | 0.2470 | 0.2543 | 83.97 |

*Table 11.* Scaling results on Qwen2.5 72B and instruction-tuned models. Qwen denotes Qwen2.5.

| Model | Method | #Params | AddSub | MultiArith | SingleEq | GSM8K | AQuA | SVAMP | Avg |
|-------|--------|---------|--------|------------|----------|-------|------|-------|-----|
| Qwen-72B (Base) | Base | — | 90.63 | 96.67 | 90.75 | 65.13 | 24.80 | 89.50 | 76.25 |
| | LoRA ($r = 16$) | 44.56M | 93.16 | 98.83 | 94.49 | 81.73 | 41.73 | 87.50 | 82.91 |
| | TopLoRA ($r = 8$) | 38.01M | 93.42 | 98.83 | 96.65 | 84.08 | 40.55 | 90.30 | 83.97 |
| | U-LoRA ($r = 8$) | 38.05M | 93.16 | 99.50 | 97.05 | 85.29 | 44.49 | 90.60 | 85.08 |
| Qwen-72B (Instruct) | Base | — | 89.37 | 97.50 | 91.73 | 71.72 | 22.83 | 88.50 | 76.94 |
| | LoRA ($r = 16$) | 44.56M | 88.35 | 98.17 | 94.69 | 85.06 | 39.76 | 85.30 | 81.89 |
| | TopLoRA ($r = 8$) | 38.01M | 88.10 | 98.67 | 97.44 | 85.14 | 41.34 | 86.40 | 82.85 |
| | U-LoRA ($r = 8$) | 38.05M | 92.91 | 99.33 | 96.46 | 85.29 | 44.09 | 86.50 | 84.17 |
| Qwen-14B (Instruct) | Base | — | 89.37 | 90.33 | 86.42 | 58.98 | 27.56 | 85.20 | 72.98 |
| | LoRA ($r = 16$) | 17.30M | 90.13 | 97.00 | 89.76 | 76.42 | 29.13 | 86.10 | 78.09 |
| | TopLoRA ($r = 8$) | 14.55M | 90.89 | 94.50 | 90.55 | 76.27 | 40.94 | 84.30 | 79.58 |
| | U-LoRA ($r = 8$) | 14.57M | 91.90 | 99.17 | 93.31 | 82.26 | 43.70 | 86.60 | 82.82 |

In comparison with TopLoRA, U-LoRA attains similar parameter counts and peak memory usage, while incurring slightly higher inference latency. This observation is consistent with a general trade-off in dynamically adaptive approaches, where finer-grained, input-dependent parameterization improves flexibility but introduces additional runtime overhead during inference. We further examine potential engineering optimizations to mitigate this overhead. Kernel fusion can reduce memory transfer overhead by combining gating computation and low-rank matrix multiplication into a single execution kernel, thereby reducing both launch and memory access costs. In addition, gating quantization can reduce computation and bandwidth pressure in the routing module with minimal impact on accuracy, as the gating signal exhibits high robustness to low-precision approximation. However, these optimizations primarily reduce constant factors and can hardly eliminate structural overhead. We clarify that the remaining extra overhead cannot be easily resolved through engineering efforts; rather, it is an inherent trade-off introduced by the dynamic, input-dependent adaptation mechanism. This predominantly stems from the inability to perform weight merging, the additional memory access caused by token-level dynamics, and sequence-level aggregation computations.

The efficiency of dynamic, non-mergeable methods such as TopLoRA and U-LoRA therefore remains an important open research direction. This is particularly relevant given that, under comparable non-mergeable inference settings, both methods exhibit similar computational characteristics, while still introducing additional overhead compared to standard LoRA due to the inherent cost of dynamic adaptation.

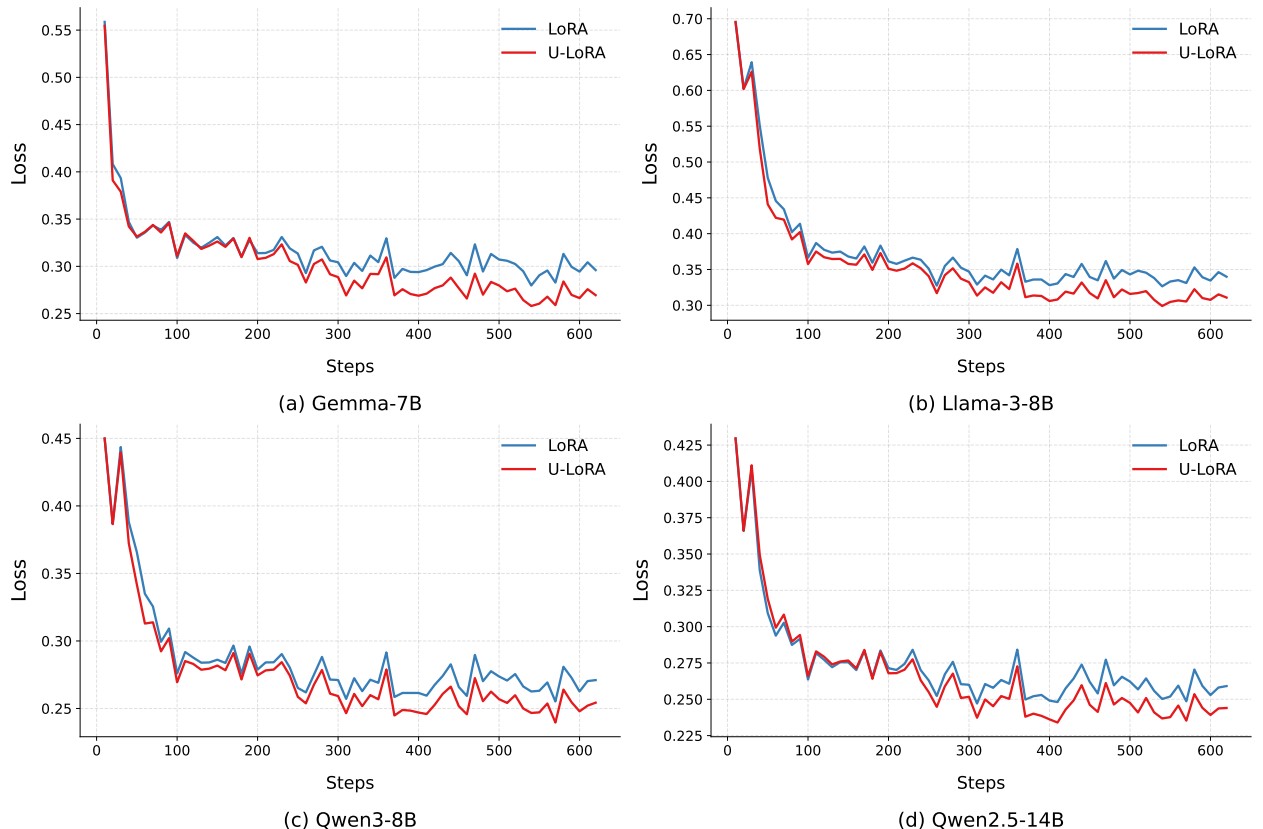

*Figure 6.* Training Loss Curves of LoRA and U-LoRA on Mathematical Reasoning Tasks.

*Table 12.* The effect of different $\beta$ values in U-LoRA on mathematical reasoning benchmarks on the Qwen3-8B model.

|  | AddSub | MultiArith | SingleEq | GSM8K | AQuA | SVAMP | Avg |
|---|---|---|---|---|---|---|---|
| $\beta = 0.9$ | 92.66 | 98.67 | 97.44 | 83.40 | 37.40 | 89.80 | 83.23 |
| $\beta = 0.95$ | 92.66 | 99.00 | 97.24 | 83.17 | 38.58 | 89.70 | 83.39 |
| $\beta = 0.98$ | 93.16 | 99.00 | 96.85 | 82.79 | 37.40 | 90.20 | 83.23 |
| $\beta = 0.99$ | 91.39 | 98.50 | 97.64 | 85.29 | 43.31 | 87.70 | 83.97 |
| $\beta = 0.995$ | 92.91 | 99.00 | 97.05 | 83.55 | 41.73 | 90.80 | 84.17 |
| $\beta = 0.998$ | 93.67 | 99.00 | 96.85 | 84.08 | 40.55 | 90.70 | 84.14 |
| $\beta = 0.999$ | 93.42 | 98.50 | 96.65 | 83.40 | 40.55 | 90.30 | 83.80 |
| $\beta = 0.9995$ | 92.91 | 99.17 | 95.47 | 84.38 | 40.94 | 87.90 | 83.46 |

*Table 13.* The standard deviation of different methods on the GLUE benchmark.

| Model | Method | RTE | MRPC | STS-B | CoLA | SST-2 | QNLI | MNLI | QQP |
|---|---|---|---|---|---|---|---|---|---|
| RoBERTa-Base | LoRA ($r = 8$) | 0.51 | 0.46 | 0.27 | 0.57 | 0.11 | 0.15 | 0.11 | 0.10 |
| | LoRA ($r = 16$) | 0.84 | 0.40 | 0.34 | 0.64 | 0.29 | 0.11 | 0.04 | 0.08 |
| | LoRA ($r = 32$) | 0.78 | 0.35 | 0.39 | 0.56 | 0.14 | 1.00 | 0.11 | 0.03 |
| | DoRA ($r = 16$) | 0.95 | 0.46 | 0.31 | 0.57 | 0.14 | 0.02 | 0.24 | 0.11 |
| | MELoRA ($r = 16$) | 0.96 | 0.64 | 0.62 | 0.52 | 0.29 | 0.14 | 0.14 | 0.03 |
| | HydraLoRA ($r = 8$) | 0.59 | 0.87 | 0.75 | 0.61 | 0.34 | 0.17 | 0.21 | 0.08 |
| | TopLoRA ($r = 8$) | 0.89 | 0.31 | 0.39 | 0.38 | 0.61 | 0.14 | 0.11 | 0.03 |
| | TopLoRA ($r = 16$) | 0.72 | 0.58 | 1.01 | 0.47 | 0.29 | 0.03 | 0.23 | 0.16 |
| | U-LoRA ($r = 8$) | 0.68 | 0.42 | 0.35 | 0.45 | 0.25 | 0.15 | 0.12 | 0.06 |
| | U-LoRA ($r = 16$) | 0.75 | 0.51 | 0.42 | 0.52 | 0.32 | 0.18 | 0.15 | 0.09 |
| RoBERTa-Large | LoRA ($r = 8$) | 0.55 | 0.81 | 0.26 | 0.64 | 0.50 | 0.22 | 0.23 | 0.05 |
| | LoRA ($r = 16$) | 0.51 | 0.20 | 0.59 | 0.98 | 0.19 | 0.16 | 0.27 | 0.04 |
| | LoRA ($r = 32$) | 0.40 | 0.66 | 0.13 | 0.65 | 0.33 | 0.44 | 0.32 | 0.02 |
| | DoRA ($r = 16$) | 0.48 | 0.20 | 0.13 | 0.56 | 0.14 | 0.33 | 0.13 | 0.07 |
| | MELoRA ($r = 16$) | 0.45 | 0.35 | 0.55 | 0.64 | 0.34 | 0.13 | 0.25 | 0.04 |
| | HydraLoRA ($r = 8$) | 0.67 | 0.72 | 0.21 | 0.29 | 0.09 | 0.09 | 0.40 | 0.09 |
| | TopLoRA ($r = 8$) | 0.90 | 0.31 | 0.07 | 0.16 | 0.09 | 0.35 | 0.15 | 0.04 |
| | TopLoRA ($r = 16$) | 0.34 | 0.58 | 0.14 | 0.78 | 0.14 | 0.12 | 0.25 | 0.10 |
| | U-LoRA ($r = 8$) | 0.62 | 0.45 | 0.18 | 0.40 | 0.22 | 0.25 | 0.22 | 0.06 |
| | U-LoRA ($r = 16$) | 0.71 | 0.55 | 0.25 | 0.58 | 0.28 | 0.30 | 0.28 | 0.08 |

*Table 14.* The standard deviation of the average accuracy on mathematical reasoning tasks.

| Method | Gemma-7B | LLaMA-3-8B | Qwen3-8B | Qwen2.5-14B |
|---|---|---|---|---|
| LoRA ($r = 8$) | 0.19 | 0.88 | 0.42 | 0.38 |
| LoRA ($r = 16$) | 0.61 | 0.62 | 0.36 | 0.34 |
| LoRA ($r = 32$) | 0.48 | 0.80 | 0.41 | 0.39 |
| DoRA ($r = 16$) | 0.58 | 0.25 | 0.22 | 0.21 |
| MELoRA ($r = 16$) | 0.31 | 0.67 | 0.28 | 0.26 |
| HydraLoRA ($r = 8$) | 0.63 | 0.39 | 0.24 | 0.20 |
| TopLoRA ($r = 8$) | 0.33 | 0.30 | 0.27 | 0.25 |
| TopLoRA ($r = 16$) | 0.52 | 0.46 | 0.34 | 0.32 |
| U-LoRA ($r = 8$) | 0.39 | 0.63 | 0.31 | 0.28 |
| U-LoRA ($r = 16$) | 0.50 | 0.48 | 0.29 | 0.27 |

