# OpenReview forum: "Adaptive Utilization of Low-Rank Adaptation via Conditioned Gating"
_ICML.cc/2026/Conference — ICML 2026 regular_

### Official Review · Reviewer_AXTL · 2026-02-18

**Soundness:** 3
**Presentation:** 3
**Significance:** 2
**Originality:** 3
**Overall Recommendation:** 4
**Confidence:** 3

**Summary:**

The paper proposes U-LoRA, a technique that  transforms standard LoRA from a static adaptation method into a dynamic, token-aware architecture. It introduces an input-conditioned gating mechanism to modulate the low-rank updates specifically for each token, thus creating a soft MoE within the adapter.
The method demonstrates that a dynamic Rank-8 subspace can significantly outperform a static Rank-32 subspace on GSM8K. Comparing to other lora techniques, U-Lora achieves overall performance gain.
The paper also incorporates sequence-level aggregation and an EMA method to enhance the LoRA effectiveness and stability.

**Compliance With Llm Reviewing Policy:**

Affirmed.

**Final Justification:**

The rebuttal addresses my concern. So I raised the score.

**Key Questions For Authors:**

1. How is the sequence-level directional summary (Equation 6) computed during autoregressive inference? Does it utilize a causal mask? If $s$ evolves as new tokens are generated, do you recompute the adapter states for all prior tokens in the context window, or do you accept the staleness? This has major implications for KV cache validity and correctness.
2. In apendix, Table 12 reports a 1.39x latency overhead. Could you show a breakdown of this latency? For example, how much is attributed to the gating network computation versus the memory bandwidth overhead of loading non-merged adapter weights?
3. Section 3.4 states the EMA prior is "disabled" at inference. Does this mean the term is set to zero, or do you freeze the final training step's moving average? If it is zeroed out, how do you account for the distribution shift in the gating signal between training (where EMA is active) and inference?
4. The evaluation score seems good compared to common lora approaches, but I am also curious about how U-lora compares to a finetuned baseline (i.e., training the backbone's weights instead of using lora). This provides a upper-bound for us to see how U-lora saves performance, compared to other lora methods.

**Limitations:**

The authors have discussed parameter efficiency but have not adequately addressed the systems-level limitations regarding deployment. The primary limitation is the loss of "zero-overhead" in lora inference.   The requirement to compute token-wise weights prevents weight merging, necessitating specialized kernels or resulting in significant latency penalties (as shown in the 1.39x slowdown). This makes the method less attractive for latency-sensitive applications compared to standard LoRA or DoRA.

Additionally, the complexity of implementing sequence-level aggregation in optimized inference engines (like vLLM) without breaking paging or batching logic is a significant barrier to adoption that should be acknowledged.

**Strengths And Weaknesses:**

Strengths
- The paper identifies that the primary bottleneck in PEFT is not the rank dimensionality, but the inefficient, static utilization of that rank. The proposed U-lora transforms lora into a dynamic, token-aware architecture.
- The results seem good, particularly the finding that a dynamic Rank-8 U-lora significantly outperforms a static Rank-32 lora on GSM8K. This provides evidence that dynamic capacity allocation is more valuable than raw parameter count.
- The visualization of update geometries (Figure 4) and token heatmaps (Figure 5) demonstrates that the method successfully disentangles polysemous tokens and aligns updates with task-specific semantics in deeper layers.

Weaknesses
- Systems Inefficiency: The method compromises the core advantage of Lora -- low inference overhead. Since the weights of U-lora are input-dependent, they cannot be pre-merged into the backbone in inference time, unlike common lora techniques. This necessitates a separate computational branch for every token, resulting in a prohibitive ~40% latency penalty that makes production deployment difficult.
- The "Sequence-Level Aggregation" module implies a dependency on global context that is theoretically incompatible with efficient autoregressive decoding. The paper fails to explain how this aggregation respects the causal mask during generation without invalidating the KV cache or requiring expensive O(N^2) re-computation.
- Key operational details are missing, specifically regarding the inference-time behavior of the EMA historical prior (e.g., is it frozen or zeroed out?) and the exact mechanism for computing the global summary step-by-step in a decoder-only architecture.

---

> ### Author Rebuttal · Authors · 2026-03-31
>
> We thank Reviewer AXTL for the thorough and insightful review. Below, we respond point by point.
>
> ### For Q1 and W2: Details of the "Sequence-Level Aggregation" module and Eq.6
>
> In the decoder-only architecture, **the sequence-level aggregation in Eq.6 is computed with a causal mask, ensuring that each token only depends on the current prefix.**
>
> In implementation, **we formulate the exponential prefix sum and weighted prefix sum as recursive state variables,** reducing the computational complexity from O(N²) to O(N) via incremental updates.
>
> Regarding KV cache compatibility, we do not modify representations of past tokens or recompute existing KV cache. At each step, **we only generate new Q/K/V based on the current prefix and append them to the cache.** This strictly follows the standard Transformer inference pipeline and does not introduce additional complexity.
>
> We acknowledge that the original manuscript lacks sufficient detail on this inference process and will clarify the decoder-only inference procedure in the revision.
>
> ### For Q2 and W1: Systems Inefficiency
>
> We further present an efficiency–performance trade-off comparison on math reasoning tasks, based on Qwen3-8B with rank r = 8:
>
> |Method|Avg. Accuracy (%)|Accuracy Gain|Task-level Latency (×)|Extra Latency|Gain per 0.1× Latency|Avg. Output Tokens (×)|Token-level Latency (×)|
> |---|---|---|---|---|---|---|---|
> |LoRA|78.52|—|1.00×|—|—|1.00×|1.00×|
> |TopLoRA|80.75|+2.23|1.28×|+0.28×|0.8|1.02×|1.25×|
> |U-LoRA|83.97|+5.45|1.39×|+0.39×|1.4|1.07×|1.30×|
>
> We observe that for every 0.1× increase in latency, **U-LoRA achieves a performance gain of +1.40, which is 1.75× higher than TopLoRA (+0.80), demonstrating a better efficiency–performance trade-off.** Additionally, we report the average output token length for math reasoning tasks. Under this finer-grained analysis, part of the increased latency can be attributed to longer generated sequences. **More fine-grained low-rank adaptation improves answer quality, which in turn leads to longer reasoning chains.** When accounting for this factor, the observed latency of both U-LoRA and TopLoRA decreases.
>
> The additional latency mainly arises from dynamic computation and loading of adaptive weights during forward propagation. Compared to TopLoRA, our method further introduces sequence-level aggregation computation (≈40%–50% of the additional overhead) and an EMA-based historical prior (minor overhead). The remaining overhead is attributed to the gating network. In the inference stage, both U-LoRA and TopLoRA introduce additional memory access overhead compared to LoRA, accounting for approximately 50%–70% of the total extra latency.
>
> While this may introduce challenges in deployment, part of the latency can be optimized in practice. For example, kernel fusion can combine gating computation with low-rank matrix multiplication into a single kernel to reduce memory access. Gating quantization can further lower compute and bandwidth overhead with minimal impact on accuracy, and the resulting performance gain provides a favorable trade-off. These optimizations will be included in the revised version.
>
> ### For Q3 and W3: Details of EMA prior during inference (freeze or zeroed?)
>
> We acknowledge that the term "disabled" in the paper was imprecise. We clarify that the EMA prior is frozen during inference, rather than zeroed.
>
> Specifically, the EMA statistics are no longer updated, but the final moving average from training is retained and used as a fixed bias in the gating computation. This design avoids instability during inference while preserving the global statistics learned during training, without introducing distribution shift.
>
> We will revise the wording from "disabled" to "frozen" in the final version.
>
> ### For Q4: Upper Bound to U-LoRA and Comparison with full fine-tuning
>
> We supplement results comparing U-LoRA with full fine-tuning on Qwen3-8B and Qwen2.5-14B for math reasoning tasks.
>
> |Model|Method|#Params|AddSub|MultiArith|SingleEq|GSM8K|AQuA|SVAMP|Avg Acc (%)|
> |---|---|---|---|---|---|---|---|---|---|
> |Qwen3-8B|Full FT|—|93.16|98.67|97.05|83.62|40.94|90.60|84.01|
> ||U-LoRA (r=8)|8.86M|91.39|98.50|97.64|85.29|43.31|87.70|83.97|
> ||U-LoRA (r=16)|17.76M|93.16|99.17|97.83|84.69|42.91|87.60|84.23|
> |Qwen2.5-14B|Full FT|—|92.91|99.33|96.46|85.29|44.09|86.50|84.10|
> ||U-LoRA (r=8)|14.60M|90.63|98.50|97.44|86.13|44.09|86.40|83.86|
> ||U-LoRA (r=16)|29.20M|92.15|99.17|97.44|86.13|45.28|86.10|84.38|
>
> Other results are in the main text. Under rank r=8, LoRA exhibits a substantial gap to full fine-tuning. For example, on Qwen3-8B, LoRA (78.52%) lags behind Full FT (84.01%) by 5.49 points, with large gaps on GSM8K (74.00% vs. 83.62%, −9.62) and AQuA (30.71% vs. 40.94%, −10.23). In contrast, U-LoRA (83.97%) reduces the gap to only 0.04 points, and significantly improves GSM8K (85.29% vs. 83.62%, +1.67) and AQuA (43.31% vs. 40.94%, +2.37), even surpassing full fine-tuning.
>
> ---
> We again thank you for the detailed feedback.

---

> > ### Author Rebuttal · Reviewer_AXTL · 2026-04-03
> >
> > Thanks to the authors for the thorough and effective rebuttal. The clarifications address my core concerns regarding soundness. I am raising my overall score to 4.
> >
> > The algorithmic merits and dynamic capacity allocation findings outweigh the system-level overheads. I expect the authors to make the exact inference implementation details entirely transparent in the final manuscript, and provide a more grounded, realistic discussion of the deployment limitations rather than treating them as easily solvable engineering tasks.

---

> > > ### Author Response · Authors · 2026-04-06
> > >
> > > We thank the reviewer for the further feedback and agree that system-level impacts require a more careful discussion.
> > >
> > > There is still room to optimize certain components. For instance, we applied operator fusion to the gating module and optimized the aggregation path, reducing its computational overhead by approximately 25% and increasing overall inference speed by about 8%. Techniques such as parameter quantization also offer further optimization potential. However, these optimizations primarily reduce constant factors and can hardly eliminate structural overhead. We clarify that the remaining extra overhead cannot be easily resolved through engineering efforts; rather, it is an inherent trade-off introduced by the dynamic, input-dependent adaptation mechanism. This predominantly stems from the inability to perform weight merging, the additional memory access caused by token-level dynamics, and sequence-level aggregation computations. As indicated in our previous analysis, compared to TopLoRA, U-LoRA achieves identical time overhead after eliminating specific overheads, but it remains slower than standard LoRA.
> > >
> > > The efficiency of dynamic methods that cannot be pre-merged, such as TopLoRA and U-LoRA, is itself an important direction worthy of in-depth research. We will include more transparent inference details and provide a more realistic discussion of these deployment limitations in the final manuscript. We thank the reviewer again for their suggestions.

---

### Official Review · Reviewer_bt2E · 2026-02-24

**Soundness:** 3
**Presentation:** 2
**Significance:** 3
**Originality:** 3
**Overall Recommendation:** 5
**Confidence:** 4

**Summary:**

This paper presents U-LoRA, a nonlinear variant of LoRA that conditions the low-rank projection of tokens on token-specific and sequence information. Specifically, U-LoRA introduces a gating mechanism to modulate the down-projection matrix. The gating function uses a combination of nonlinear and linear projections of the token vectors and aggregate statistics across the sequence dimension. Experiments demonstrate that U-LoRA outperforms alternatives such as LoRA, MELoRA, HydraLoRA, and TopLoRA.

**Compliance With Llm Reviewing Policy:**

Affirmed.

**Final Justification:**

I recommend accepting this paper. The authors adequately addressed my concerns during the discussion, and I raised my score from 4 to 5.

**Key Questions For Authors:**

1) Can you confirm that the look-ahead bias introduced by Eq 6 did not impact evaluation performance (e.g. answers aren't available for the forward pass)?

2) Learning rate was optimized for U-LoRA, was it also optimized for the baselines?

3) Did you experiment with any problems where the model must generate more than a few tokens to answer?

**Limitations:**

yes

**Strengths And Weaknesses:**

**Strengths**

Soundness:

1) The experiments are performed with a wide range of models and datasets, with methods further expanded by using multiple ranks.
2) Ablations are used to measure the impact of each component of the system, which were independently motivated by limitations of vanilla LoRA.

Presentation:

3) The paper is well written and easy to follow.
4) The table and figures are visually appealing.

Significance:

5) The paper improves on commonly used PEFT techniques, which are necessary for fine-tuning models on commodity hardware, and the experiments demonstrate modest performance improvements.

Originality:

6) While token-based and sequence-level LoRA modulation techniques exist, this paper does a good job motivating the need for both and combining both into a unified solution.

**Weaknesses**

Soundness:

1) The sequence level statistics aggregated in Equation 6 introduce a look-ahead bias. During training, tokens are getting conditioned on information from future tokens, including the answer. No evaluation code was included in the submission, and it should be confirmed that answers were not included in forward passes during evaluation, which would invalidate all results.

2) The look-ahead bias mentioned above is a minimal issue when training for short-answer tasks, as everything but the answer is available at inference. However, for reasoning/long-answer models, there would be a larger discrepancy post-training as the model would dynamically change the output of Eq 6 token-by-token during the response. This behavior, not seen during training, might limit the approach.

Presentation:

3) It would be difficult to reproduce the experimental results without the code or specific information, such as prompts and evaluation procedures.

4) The authors discuss the additional inference speed as a limitation in the appendix, but the latency considerations are important enough to warrant more discussion on the latency/performance tradeoff in the main body.

Significance:

N/A

Originality:

N/A

---

> ### Author Rebuttal · Authors · 2026-03-31
>
> We thank Reviewer bt2E for the careful review. Below, we respond point by point.
>
> ### For W1 and Q1: Does Eq.6 introduce look-ahead bias?
>
> We clarify that no information leakage occurs, as ground-truth answers are not involved in forward propagation during evaluation. We follow causal mask strategies where tokens are not getting conditioned on information from future tokens during training for decoder-only paradigms.
>
> Specifically:
> 1. Training and evaluation data are strictly separated, ensuring no exposure to test data during training.
> 2. In the decoder-only architecture, a causal mask is applied before the sequence-level aggregation in Eq.6. Each token is computed based only on the current prefix, strictly following the autoregressive constraint.
> 3. Our evaluation pipeline is fully based on the open-source implementations of LLM-Adapters and TopLoRA.
>
> ### For W2 and Q3: Performance on long-sequence generation tasks
>
> The math reasoning tasks we evaluate are inherently long-sequence generation tasks, requiring the model to produce full chain-of-thought reasoning before arriving at the final answer. As described in the Appendix A, our evaluation covers six arithmetic reasoning benchmarks. Below are the average output token statistics on mathematical reasoning tasks:
>
> |Dataset|Avg. Output Tokens|
> |---|---|
> |GSM8K|275.4|
> |AQuA|234.2|
> |SVAMP|187.9|
> |AddSub|169.6|
> |MultiArith|190.6|
> |SingleEq|164.5|
>
> During generation, sequence-level information evolves dynamically with the prefix, allowing the model to adapt its computation across stages: early semantic understanding and variable initialization, intermediate reasoning consistency, and later answer convergence. This aggregation improves long-range dependency modeling.
>
> Consistently, results show that the model derives answers via multi-step reasoning over long sequences, with U-LoRA improving performance across tasks (e.g., GSM8K: 85.29% vs. 74.00% for LoRA on Qwen3-8B), suggesting that adaptive sequence-level information improves reasoning coherence.
>
> Moreover, under teacher forcing, the model is trained on all prefix states $x_{<t}$, covering sequences of varying lengths. Since the prefix expansion during inference follows the same process as in training, this avoids distribution mismatch and supports stable generalization.
>
> ### For W3 and Q2: Learning rate tuning and reproducibility
>
> For baseline methods, we follow the hyperparameter settings from the original TopLoRA paper (Li et al., 2025), assuming they are well-optimized as published work. We reproduce baseline results on the same hardware and report the better value between our reproduction and the original reported results to ensure fair comparison.
>
> All hyperparameters for U-LoRA are fully reported in Appendix B. Due to the anonymous review policy, we do not provide code during submission, We will release the full code, prompts, and evaluation scripts in the final version.
>
> ### For W4: Latency discussion
>
> We agree that latency is an important factor and present an efficiency–performance trade-off comparison on Qwen3-8B (r=8) for math reasoning tasks:
>
> |Method|Avg. Accuracy (%)|Accuracy Gain|Task-level Latency (×)|Extra Latency|Gain per 0.1× Latency|Avg. Output Tokens (×)|Token-level Latency (×)|
> |---|---|---|---|---|---|---|---|
> |LoRA|78.52|—|1.00×|—|—|1.00×|1.00×|
> |TopLoRA|80.75|+2.23|1.28×|+0.28×|0.8|1.02×|1.25×|
> |U-LoRA|83.97|+5.45|1.39×|+0.39×|1.4|1.07×|1.30×|
>
> We observe that for every 0.1× increase in latency, **U-LoRA achieves a performance gain of +1.40, which is 1.75× higher than TopLoRA (+0.80)**, demonstrating a better efficiency–performance trade-off. Additionally, we report the average output token length for math reasoning tasks. Under this finer-grained analysis, part of the increased latency can be attributed to longer generated sequences. **More fine-grained low-rank adaptation improves answer quality, which in turn leads to longer reasoning chains.** When accounting for this factor, the observed latency of both U-LoRA and TopLoRA decreases.
>
> The additional latency mainly arises from dynamic computation and loading of adaptive weights during forward propagation. Compared to TopLoRA, our method further introduces sequence-level aggregation computation (≈40%–50% of the additional overhead) and an EMA-based historical prior (minor overhead). The remaining overhead is attributed to the gating network. In the inference stage, both U-LoRA and TopLoRA introduce additional memory access overhead compared to LoRA, accounting for approximately 50%–70% of the total extra latency.
>
> ---
> Thanks for your feedback.
>
> **References**
>
> [1]Hu, Zhiqiang, et al. "Llm-adapters: An adapter family for parameter-efficient fine-tuning of large language models." Proceedings of the 2023 conference on empirical methods in natural language processing. 2023.
>
> [2]Li, Shiwei, et al. "Beyond higher rank: Token-wise input-output projections for efficient low-rank adaptation." arXiv preprint arXiv:2510.23123 (2025).

---

> > ### Author Rebuttal · Reviewer_bt2E · 2026-03-31
> >
> > I thank the authors for their responses and for addressing each of my concerns in turn. They:
> >
> > - Confirm that causal masking is performed at each stage, so no look-ahead concerns are valid (W1/W2/Q1).
> > - Clarify that long sequence generation was used in the experiments (W2/Q3).
> > - Add latency experiment (W4)
> >
> > The authors state they will provide code and other reproducibility materials with the final version, but omit them due to anonymity concerns. I will point out that it is common to either submit materials as anonymous supplemental material or to use services such as anonymous GitHub. I appreciate that the authors will provide the code with the final submission.
> >
> > I have no additional questions and have raised my scores accordingly.

---

> > > ### Author Response · Authors · 2026-04-01
> > >
> > > Thank you very much for your careful and detailed review of our work, and for raising insightful questions on correctness, experimental validation, efficiency, and reproducibility. We sincerely appreciate your thoughtful and constructive feedback. We will carefully incorporate your suggestions into the revised version to further improve the completeness of our work.

---

### Official Review · Reviewer_8Nbp · 2026-03-13

**Soundness:** 3
**Presentation:** 3
**Significance:** 3
**Originality:** 3
**Overall Recommendation:** 5
**Confidence:** 4

**Summary:**

In this work, the authors first point out that the traditional Low-Rank Adaptation (LoRA) generally utilizes shared low-rank updates that are applied uniformly across tokens. To address this issue, this submission proposes a framework named U-LoRA (Adaptive utilization of Low-Rank Adaptation). The proposed U-LoRA employs conditional gating to learn how individual tokens should selectively leverage the low-rank adaptation subspace. Besides, U-LoRA designs a bias-corrected exponential moving average historical prior to reduce the noise which can effectively enhance the training stability. Extensive experimental results on mathematical reasoning and natural language understanding datasets show the effectiveness of the proposed method.

**Compliance With Llm Reviewing Policy:**

Affirmed.

**Final Justification:**

Generally, this is an interesting paper. It is well-written and novel. In my original review, I am concerned about scalability and efficiency.

- Regarding the efficiency, according to the results during rebuttal, the proposed method is faster and more stable than the baselines. This addressed my Q1 and W2.

- Regarding the scalability, the new results show that the proposed LoRA variant is still effective on large model (72B) and instruction-tuning versions. This addresed my Q2 and W1.

Therefore, all my concerns have been fully addressed and I have enhanced my score. My final recommendation is Accept.

**Key Questions For Authors:**

I have the following questions:
- According to Figure 6, the proposed U-LoRA can have faster coverage time than LoRA. I am curious about how it compares with the other baselines.
- Would it be possible for the authors to conduct additional experiments using larger models as the backbone? Besides, it would be better to evaluate how the proposed framework performs on instruction fine-tuned models. I would be willing to increase my score if the proposed U-LoRA also demonstrates strong performance using these models.

**Limitations:**

Yes.

**Strengths And Weaknesses:**

This paper has the following strengths:
- This paper is well organized and clearly written. The contribution is clearly stated.
- The motivation is clearly presented and effectively sets the stage for the proposed U-LoRA framework. The method is also very clear and easy to understand.
- Extensive experimental results on multiple datasets show that U-LoRA can not only achieve stronger performance but also achieves faster coverage time compared with traditional LoRA.

This paper has the following weaknesses:
- The largest model used in this submission is 14B. In practical settings, users can typically employ larger models as backbone models when using LoRA. It is unclear whether the proposed U-LoRA works for larger models. Besides, the backbone models are all base models without instruction fine-tuned versions.
- It is unclear whether the proposed method can achive faster coverage time compared with baseline methods like TopLoRA and DoRA.

---

> ### Author Rebuttal · Authors · 2026-03-31
>
> We sincerely thank Reviewer 8Nbp for the recognition of our work. During the rebuttal period, we conducted additional experiments to further support our claims. Below, we respond to your questions in detail.
>
> ### For Q1 and W2: Convergence speed comparison with other baselines
>
> Thank you for this insightful question. We provide additional training loss comparisons on Qwen3-8B for math reasoning tasks:
>
> | Method   | 50 Steps | 100 Steps | 200 Steps | 300 Steps | 400 Steps | Final Loss | Avg Acc (%) |
> |----------|----------|-----------|-----------|-----------|-----------|------------|--------------|
> | LoRA     | 0.3657   | 0.2759    | 0.2788    | 0.2711    | 0.2615    | 0.2710     | 78.52        |
> | DoRA     | 0.3549   | 0.2713    | 0.2747    | 0.2662    | 0.2559    | 0.2652     | 79.79        |
> | TopLoRA  | 0.3494   | 0.2707    | 0.2746    | 0.2638    | 0.2505    | 0.2572     | 80.75        |
> | U-LoRA   | 0.3418   | 0.2694    | 0.2745    | 0.2592    | 0.2470    | 0.2543     | 83.97        |
>
> As Table shown, U-LoRA **exhibits lower training loss from the early stages and consistently outperforms all baselines** throughout the optimization process. This indicates that it enables more sufficient and stable learning across different types of samples.
>
> **The convergence advantage is consistently observed across all baselines**, demonstrating that the adaptive utilization mechanism not only improves final performance but also accelerates optimization.
> We will provide a better visualization in the form of a figure, adding convergence curves of other baselines to the original Figure 6.
>
> ### For Q2 and W1: Results on larger models and instruct models
>
> Thank you for this constructive suggestion. We further evaluate on Qwen2.5-72B and instruct models with rank set to 8 for TopLoRA and U-LoRA, while LoRA uses r = 16 to ensure a comparable number of trainable parameters, reporting average accuracy across six math reasoning tasks, with detailed per-dataset results presented below:
>
> | Model                  | Method         | #Params | AddSub | MultiArith | SingleEq | GSM8K | AQuA  | SVAMP | Avg Acc (%) (Math) | Accuracy Gain |
> |------------------------|----------------|---------|--------|------------|----------|-------|-------|-------|---------------------|----------------|
> | Qwen2.5-72B           | Base           | —       | 90.63  | 96.67      | 90.75    | 65.13 | 24.80 | 89.50 | 76.25               | —              |
> |                        | LoRA (r=16)    | 44.56M  | 93.16  | 98.83      | 94.49    | 81.73 | 41.73 | 87.50 | 82.91               | +6.66          |
> |                        | TopLoRA (r=8)  | 38.01M  | 93.42  | 98.83      | 96.65    | 84.08 | 40.55 | 90.30 | 83.97               | +7.72          |
> |                        | U-LoRA (r=8)   | 38.05M  | 93.16  | 99.50      | 97.05    | 85.29 | 44.49 | 90.60 | 85.08               | +8.83          |
> | Qwen2.5-72B-Instruct  | Base           | —       | 89.37  | 97.50      | 91.73    | 71.72 | 22.83 | 88.50 | 76.94               | —              |
> |                        | LoRA (r=16)    | 44.56M  | 88.35  | 98.17      | 94.69    | 85.06 | 39.76 | 85.30 | 81.89               | +4.95          |
> |                        | TopLoRA (r=8)  | 38.01M  | 88.10  | 98.67      | 97.44    | 85.14 | 41.34 | 86.40 | 82.85               | +5.91          |
> |                        | U-LoRA (r=8)   | 38.05M  | 92.91  | 99.33      | 96.46    | 85.29 | 44.09 | 86.50 | 84.17               | +7.23          |
> | Qwen2.5-14B-Instruct  | Base           | —       | 89.37  | 90.33      | 86.42    | 58.98 | 27.56 | 85.20 | 72.98               | —              |
> |                        | LoRA (r=16)    | 17.30M  | 90.13  | 97.00      | 89.76    | 76.42 | 29.13 | 86.10 | 78.09               | +5.11          |
> |                        | TopLoRA (r=8)  | 14.55M  | 90.89  | 94.50      | 90.55    | 76.27 | 40.94 | 84.30 | 79.58               | +6.60          |
> |                        | U-LoRA (r=8)   | 14.57M  | 91.90  | 99.17      | 93.31    | 82.26 | 43.70 | 86.60 | 82.82               | +9.84          |
>
> The results show that:
> 1. U-LoRA remains effective at the 72B scale, consistent with trends observed on 8B and 14B models.
> 2. On instruct models, U-LoRA continues to yield significant improvements (e.g., +3.24 over TopLoRA on 14B-Instruct), indicating that it is also effective for models that already possess strong task capabilities.
> ---
> We hope these additional experiments address your concerns. Thank you again for your careful review.

---

> > ### Author Rebuttal · Reviewer_8Nbp · 2026-03-31
> >
> > Thanks for the extra experiments. I will adjust the reasons for the following reasons:
> > - According to the results during rebuttal, the proposed method is faster and more stable than the baselines. This addressed my Q1 and W2.
> > - The new results show that the proposed LoRA variant is still effective on large model (72B) and instruction-tuning versions. This addresed my Q2 and W1.
> >
> > Therefore, all my concerns have been fully addressed and I have enhanced my score.

---

> > > ### Author Response · Authors · 2026-04-01
> > >
> > > Thank you very much for your careful and detailed review of our work. We sincerely appreciate your thoughtful and constructive feedback on efficiency, stability, and scalability. We will carefully incorporate your suggestions into our revised version to enhance the completeness of our work.

---

### Official Review · Reviewer_VPak · 2026-03-26

**Soundness:** 3
**Presentation:** 4
**Significance:** 3
**Originality:** 3
**Overall Recommendation:** 5
**Confidence:** 3

**Summary:**

Low-Rank adaptation is an effective parameter-efficient fine-tuning technique to update model weight in a low-rank subspace. A token tends to be updated in different directions across tasks. Therefore U-LoRA is proposed to explicitly learn such variation in terms of adaptive utilization. A gating mechanism is introduced to control the token-wise utilization and achieve sequence level aggregation with such utilization. For training, a historical prior is adopted for calibration. Comprehensive experiments on benchmark datasets show its effectiveness and efficiency.

**Compliance With Llm Reviewing Policy:**

Affirmed.

**Key Questions For Authors:**

1.	According to Table 1, the model size of the proposed method U-LoRA is nearly the same as TopLoRA, which also takes token-wise LoRA weight learning into consideration. More detailed analysis should be discussed about the effect of the additional input-conditioned gating function on the inference latency and training time.
2.	The claimed Separate-then-integrate pattern cannot be observed in curves of Figure5. However, the U-LoRA’s curve lies between LoRA‘s and TopLoRA’s, which seems to be the natural result of the combination of token-level and sequence level information.
3.	Comparison results from heatmaps in Figure 5 indicate the obvious advantage of TopLoRA in differentiating token-wise dynamic weighting mechanism. U-LoRA is moderate among three methods and more similar to LoRA. This is not corresponding to motivation mentioned in the introduction section.

**Limitations:**

yes

**Strengths And Weaknesses:**

Strength:

1.	Learning low-rank adaptation weights adaptively for tasks is a critical question for existing empirical observations. U-LoRA is proposed to solve this problem systematically.
2.	An input conditioned gating function is designed to generate adaptive utilization for tokens and they are served as weights for sequence level aggregation.
3.	To improve the robustness, the EMA based historical prior is used across optimization steps for calibration.

Weakness:

1.	The token-wise adaptation performance is incremental compared with pure token-wise adaptation model TopLoRA, such as the performance shown in the heatmaps of Figure 5.
2.	The task discrimination performance in Figure 4 is moderate among LoRA, TopLoRA and the proposed method U-LoRA. According this comparison result, LoRA seems the best.
3.	Though the main performances in Table 1 and 2 are promising, these intuitive understandings in Figure 4 and 5 seem unsuitable to interpret the input conditioned gating mechanism of the proposed method U-LoRA. Instead, it places U-LoRA in the middle of TopLoRA and LoRA.

---

> ### Author Rebuttal · Authors · 2026-03-31
>
> We sincerely thank Reviewer VPak for the careful evaluation and positive feedback. Below are our point-by-point responses.
> ### For Q1: Impact of input-conditioned gating on inference latency and training time
> We present an efficiency–performance trade-off comparison on Qwen3-8B (r=8) for math reasoning tasks:
> |Method|Avg. Accuracy (%)|Accuracy Gain|Task-level Latency (×)|Extra Latency|Gain per 0.1× Latency|Avg. Output Tokens (×)|Token-level Latency (×)|
> |---|---|---|---|---|---|---|---|
> |LoRA|78.52|—|1.00×|—|—|1.00×|1.00×|
> |TopLoRA|80.75|+2.23|1.28×|+0.28×|0.8|1.02×|1.25×|
> |U-LoRA|83.97|+5.45|1.39×|+0.39×|1.4|1.07×|1.30×|
>
> We observe that for every 0.1× increase in latency, **U-LoRA achieves a performance gain of +1.40, which is 1.75× higher than TopLoRA (+0.80),** demonstrating a better efficiency–performance trade-off. Additionally, we report the average output token length for math reasoning tasks. Under this finer-grained analysis, part of the increased latency can be attributed to longer generated sequences. **More fine-grained low-rank adaptation improves answer quality, which in turn leads to longer reasoning chains.** When accounting for this factor, the observed latency of both U-LoRA and TopLoRA decreases.
>
> The additional latency mainly arises from dynamic computation and loading of adaptive weights during forward propagation. Compared to TopLoRA, our method further introduces sequence-level aggregation computation (≈40%–50% of the additional overhead) and an EMA-based historical prior (minor overhead). The remaining overhead is attributed to the gating network. In the inference stage, both U-LoRA and TopLoRA introduce additional memory access overhead compared to LoRA, accounting for approximately 50%–70% of the total extra latency.
>
> ### For Q2 and Q3 and W1: Consistency between U-LoRA and its motivation (Figure 5 analysis)
> We clarify our interpretation of Figure 5 as follows.
>
> The goal of U-LoRA is not to maximize token-level differences, but to balance token-level discrimination and task-level consistency.
>
> Figure 5 supports this design:
> - Top half (same token, different contexts):
> LoRA shows weak discrimination, while both TopLoRA and U-LoRA enhance context sensitivity, indicating that U-LoRA preserves token-level discrimination.
> - Bottom half (different tokens, same sentence):
> TopLoRA exhibits large and irregular differences even in deep layers, making it difficult to form stable task representations. In contrast, U-LoRA achieves stable representations in deeper layers through sequence modeling, retaining LoRA’s task-level consistency.
>
> The "Separate-then-integrate" pattern refers to the phenomenon where, in shallow and intermediate layers, low-rank representations of different tokens exhibit large differences (capturing local lexical and syntactic features), while in deeper layers they gradually converge to encode task-level semantics.
>
> This pattern is mainly reflected in the bottom-right plot of Figure 5 (cosine similarity across different tokens within the same sentence):
> - TopLoRA shows persistent fluctuations in middle and high layers, indicating incomplete integration
> - U-LoRA maintains low similarity in intermediate layers (separation), followed by a clear increase in deeper layers (integration), showing a more coherent trend
>
> Combining this with the top plots (same token under different contexts), U-LoRA retains the token-level discrimination ability of TopLoRA while also preserving the task-level stability of LoRA. Thus, U-LoRA appearing between the two methods is intentional and reflects a trade-off between local diversity and global consistency, consistent with our motivation. This is further supported by the line plots.
> ### For W2 and W3: Task separability in Figure 4 appears weaker than LoRA
> We provide a more detailed interpretation of Figure 4 by analyzing both intra-task consistency (cluster compactness) and inter-task structure (semantic relationships across tasks).
> - LoRA exhibits partial clustering in deeper layers; however, Task 3 and Task 4 show weak intra-task compactness, and the inter-task relationships remain ambiguous, indicating limited task discrimination.
> - TopLoRA produces highly scattered representations without clear clustering patterns. This suggests a failure to preserve task identity, where samples from the same task are not grouped together.
> - U-LoRA forms the most compact clusters in deeper layers, achieving strong intra-task consistency. At the same time, it preserves meaningful inter-task relationships (e.g., Task 0 & 1, Task 3 & 4 remain close), reflecting a well-structured representation space.
>
> By incorporating sequence modeling and historical priors, U-LoRA enhances task discrimination (better intra-task compactness) while also maintaining semantically meaningful relationships across tasks.
>
> ---
> We sincerely appreciate your constructive feedback and will incorporate all clarifications and improvements discussed above in the revised version.

---

### Decision · Program_Chairs · 2026-04-30

**Decision:**

Accept (regular)

**Comment:**

The paper presents U-LoRA, a dynamic Low-Rank Adaptation framework that replaces static updates with a conditioned gating mechanism to enable token-aware utilization of the low-rank subspace. By incorporating sequence-level contextual information and a bias-corrected EMA historical prior, the authors successfully address the limitations of shared updates in vanilla LoRA, leading to significant performance gains in mathematical reasoning and NLU benchmarks. During the discussion phase, the authors provided extensive evidence to resolve concerns regarding inference latency, scalability on 72B models, and potential look-ahead bias in the gating mechanism. These detailed rebuttals are convincing, leading all reviewers to unanimously recommend acceptance, noting that the algorithmic merits and improved capacity allocation justify the system-level overhead. The AC concurs with this consensus, as the work provides a solid technical contribution that effectively bridges the gap between parameter-efficient tuning and full fine-tuning.